# TopGQ: Fast GNN Post-Training Quantization Leveraging Topology Information

## Abstract

Graph Neural Networks (GNNs) demand substantial memory and computation as datasets scale in size. Thus, quantization is a promising remedy by compressing full-precision values into low-bit representations. However, existing GNN quantization methods depend on tedious gradient-based updates to preserve accuracy. This quantization time may be a major barrier to real-world deployments as the input graph size scales. To this end, we present **TopGQ** (**Top**ology-aware **GNN Q**uantization), an accurate post-training quantization framework tailored for GNNs, alleviating the burden of redundant quantization overhead. We propose *Dual-axis scale absorption*, which applies scale factors along both activation axes, merging one into the static adjacency matrix. Dual-axis scale absorption attains higher accuracy via addressing outlier nodes. This helps maintain the same computational cost as column-wise quantized inference. We further introduce topology-guided quantization, which exploits the relationship between local graph structure and activation variance. TopGQ enables fast inference for unseen nodes, via a *novel node index (TopPIN)*, a lightweight proxy of activation variance from local structure. With these techniques, TopGQ eliminates the need for retraining while preserving accuracy. Experimental results show that TopGQ is comparable to prior works while reducing quantization time by an order of magnitude, establishing it as a practical solution for *efficient and scalable* GNN inference.

## 1 Introduction

Graph neural networks (GNNs) have attracted a great amount of attention due to their ability to process diverse unstructured data in diverse domains, such as recommendation systems (Zhang et al., 2023), molecular interaction (Wale et al., 2008), transportation networks (Cao et al., 2020), and social network analysis (Arazzi et al., 2023). Although model sizes are typically small (Wu et al., 2020), they introduce an extensive amount of computation and memory costs from activations, i.e., the node and edge features (Liu et al., 2021). Especially with the trend where the graph size is continuously growing (Liu et al., 2024a; Hu et al., 2020), there is a growing need to process large graphs with limited resources.

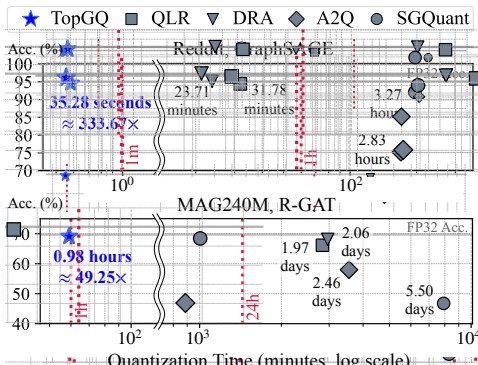

Figure 1: Quantization time-accuracy trade-off plot with large-scale graph datasets.

One promising approach to address this need is quantization, which reduces memory usage and computational costs by using low-bit representations (Ashkboos et al., 2024; Liu et al., 2024b; Li et al.; 2023). However, quantizing GNNs is known to be difficult due to the varying magnitudes of node activations. The outlier node features are induced by aggregation from the message-passing algorithm, leading to quantization errors (Tailor et al., 2020; Zhu et al., 2022; Wei et al., 2022). Accordingly, existing methods target managing these outliers.

These GNN quantization methods demand extensive parameter tuning or long training time. Quantization-aware training (QAT) methods involve costly retraining, often exceeding full-precision

training time (Feng et al., 2020; Tailor et al., 2020; Huang et al., 2022; Zhu et al., 2022; Wang et al., 2023). Post-training quantization (PTQ) methods typically avoid such overhead by keeping model weights fixed. However, the existing PTQ method (Jeddi et al., 2024) still employs gradient-based iterations on quantization parameters, negating its expected advantage. Section 1 illustrates such a phenomenon, where their quantization time for large graphs can take up much more than a day.

This large quantization time poses a major barrier to quantized GNN deployment in real-world scenarios, particularly when frequent model updates are required. Popular GNN applications such as personalization and recommendation (Wu et al., 2023; Chen et al., 2023; You et al., 2022; Guan et al., 2025) operate on large-scale graphs and benefit from quantization. However, these applications often require model updates on a minute-to-hour scale (Liu et al., 2022; Ni et al., 2021), and excessive quantization time makes deployment impractical by canceling out the benefits of quantization.

For this, we present TopGQ, a PTQ method which achieves *orders of magnitude faster* quantization with comparable or even better task performance. First, we show that node-wise quantization is more preferable for GNN (Section 4), due to the existence of outlier nodes. However, node-wise quantization on the aggregation phase prevents integer arithmetic. To this end, we propose (1) dual-axis scale absorption, a technique that enables fast and accurate integer matrix multiplication with node-wise scaling. It merges the scaling factors of each node into the adjacency matrix, preserving efficiency while maintaining accuracy. We further propose (2) TopPIN, a lightweight node index that encodes local topology to guide quantization. TopPIN enables rapid assignment of quantization parameters to unseen nodes, ensuring fast inference. We show theoretically and empirically that TopPIN is a sound proxy for unseen node assignment. Extensive experimental results validate that TopGQ outperforms current state-of-the-art baselines, achieving orders of magnitude faster quantization while preserving accuracy and inference latency, establishing a new standard in GNN quantization.

## 2 BACKGROUND

**Graph neural networks.** Let $G = (V, E)$ be a directed graph with $n = |V|$ nodes, $v_1, \ldots, v_n$. Denote $A \in \mathbb{R}^{n \times n}$ as the adjacency matrix, where $A_{ij} = 1_{(v_j, v_i) \in E}$. For node $v_i$, define its closed in-neighborhood as $\mathcal{N}(v_i) = \{ v_j \mid (v_j, v_i) \in E \} \cup \{v_i\}$, and let degree $d(v_i) = |\mathcal{N}(v_i)|$. We denote $D = \text{diag}(d(v_1), d(v_2), \ldots, d(v_n))$ as the diagonal degree matrix, $h_i$ as feature vector of $v_i$.

To embed topology, GNNs aggregate information from neighboring nodes $v_j \in \mathcal{N}(v_i)$ to update $h_i$. This procedure is referred to as the *message-passing* algorithm, which consists of two steps: *combination* and *aggregation*. First, the hidden node feature $h_i^{(l)}$ is multiplied by the weight matrix $W^{(l)}$ in the $l$-th GNN layer. Next, the feature is aggregated to $h_i^{(l+1)}$ as follows:

$$h_i^{(l+1)} = \phi\Big(\bigoplus_{\{j \mid v_j \in \mathcal{N}(v_i)\}} W^{(l)} h_j^{(l)}\Big), \tag{1}$$

where $\phi$ is an update function, and $\bigoplus$ is a permutation-invariant operator such as *sum* or *mean*.

The GNN computation can also be formulated in matrix form. Let $X^{(l)} = [h_1^{(l)}, \ldots, h_n^{(l)}]^T \in \mathbb{R}^{n \times d_l}$ be the matrix of node features at layer $l$, and let $W^{(l)} \in \mathbb{R}^{d_l \times d_{l+1}}$ be the weight matrix. Then, using an augmented adjacency matrix $\tilde{A} \in \mathbb{R}^{n \times n}$, the combination and aggregation steps are:

$$X_c^{(l)} = X^{(l)}W^{(l)}, \quad X^{(l+1)} = \sigma\big(\tilde{A} X_c^{(l)}\big), \tag{2}$$

where $\sigma$ is a nonlinear function. The specific form of $\tilde{A}$ varies by GNN architecture. For example, GCN (Kipf & Welling, 2016) employs the normalized graph Laplacian $\tilde{A} = D^{-1/2}(A + I_n)D^{-1/2}$, while GIN (Xu et al., 2019) uses the binary matrix $\tilde{A} = A + I_n$. GraphSAGE (Hamilton et al., 2017) differs by sampling a subset of neighbors instead of using the entire neighborhood at aggregation.

**Transductive and inductive settings.** GNN training operates in either a transductive or an inductive setting. In the transductive setting, the entire graph (e.g., features and topology) is available during training, except for the test node labels. As a result, inference can be done with precomputed embeddings (Xu et al., 2024), leaving little room for acceleration and quantization benefits. In contrast, the inductive setting introduces unseen nodes or graphs at test time, requiring computation of node embedding during inference. Consequently, GNN quantization would have a much greater impact in inductive settings, where reducing computation and memory directly accelerates

inference. Moreover, the inductive setting better reflects real-world scenarios where graphs evolve or differ from those used for training—such as social networks with new users, recommendation systems between users and new items, or molecular property prediction for unseen molecules—and is therefore generally considered the more practical and deployment-oriented evaluation setting.

**Quantization** replaces high-precision floating-point operations with low-bit integer operations, thereby reducing computational cost and memory usage. We adopt uniform integer quantization with scale ($s$) and zero-point ($z$). Given a tensor $X$, each element $x \in X$ is quantized as:

$$x^q = Q(x; s, z) = \text{clamp}\Big(\Big\lfloor \frac{1}{s} \cdot (x - z) \Big\rceil, q_{\min}, q_{\max}\Big), \quad s = \frac{x_{\max} - x_{\min}}{q_{\max} - q_{\min}}, \quad (3)$$

$q_{\min}$ and $q_{\max}$ are the minimum and maximum integer values in $k$-bit representation, and $\lfloor \cdot \rceil$ denotes rounding. Quantization operates at various granularities, such as per-tensor, per-column, or per-row. Quantization typically follows either post-training quantization (PTQ) or quantization-aware training (QAT). QAT iteratively updates the weights using the calculated gradients, whereas PTQ calibrates scale and zero-point without updating model weights, and therefore much faster in general.

## 3 RELATED WORK

**GNN quantization** efficiently reduce extensive memory and computational costs of GNNs (Kipf & Welling, 2016; Veličković et al., 2018; Xu et al., 2019; Hamilton et al., 2017). Degree-Quant (Tailor et al., 2020) is the first work to quantize GNN using QAT, excluding high-degree node activations in calibration for robust quantization parameters and compressing later at inference. EPQuant (Huang et al., 2022) utilizes product quantization for reducing the high memory cost. SGQuant (Feng et al., 2020) and $A^2Q$ (Zhu et al., 2022) are also QAT methods, but they differ in that they allow mixed-precision to assign a higher bitwidth to high-magnitude features. The quantization parameters are optimized with gradients in QLR (Wang et al., 2023) and DRA (Jeddi et al., 2024). While QLR leverages these parameters with customized message propagation, DRA optimizes them to reconstruct the FP32 distributions. Thus, they require significant and redundant quantization overheads, whereas TopGQ allows orders of magnitude shorter quantization time (Section 1).

**Graph topology in GNNs** is often integrated during training to help the model effectively learn the structural information (Ji, 2019; Zhang & Lu, 2020; Hu et al., 2022; Wu et al., 2018; You et al., 2021; Brasoveanu et al., 2023). For example, (Ji, 2019) uses degree centrality to find highly central nodes for effective representation learning. Also, (Zhang & Lu, 2020) uses betweenness centrality to assign weights to each node during aggregation. There are prior attempts to leverage topology for binarization of graph neural networks (Bahri et al., 2021; Jing et al., 2021). However, these methods do not incorporate topology in relation to per-node activation statistics for GNN quantization.

## 4 TOPOLOGY-AWARE GNN QUANTIZATION: NECESSITY AND CHALLENGES

**Necessity of topology-aware GNN quantization.** GNN quantization requires special consideration due to its unique message-passing mechanism. In particular, the accumulation of neighborhood information induces substantial diversity across nodes, making node-wise quantization a preferred approach. Figure 2 illustrates such behavior by comparing the range of values within each node dimension (Figures 2a and 2c) and feature dimension (Figures 2b and 2d). Figures 2a and 2c presents that node-wide ranges are more concentrated, with high similarity between the 5th–95th percentile range and the min–max range. This indicates the absence of extreme outliers within each node group, making it favorable for quantization. However, in the feature-wise plots (Figures 2b and 2d), each min-max range is much broader, while 95% of the values exist within a much narrow interval. This distribution is more prone to outliers, leading to wasted quantization bins and higher error. This indicates that node-wise quantization is a more favorable choice for the activation in GNNs.

Based on the observation, we assign different quantization scales to the group of nodes for the feature matrix $X$ in both the combination and the aggregation phase of GNN inference. Enabling such a method in the *combination* phase is relatively straightforward. In fact, existing methods (Zhu et al., 2022; Feng et al., 2020) already employ node-wise quantization:

$$X \cdot W \approx \text{diag}(S_X) \cdot X^Q \cdot W^Q \cdot \text{diag}(S_W) = (S_X \cdot S_W^\top) \odot (X^Q \cdot W^Q) \quad (4)$$

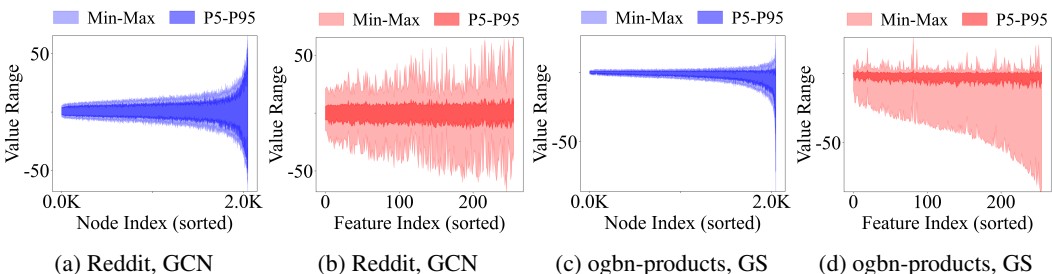

Figure 2: Node-wise and feature-wise range plot, sorted in ascending order. 'Node Index' indicates each node, and 'Feature Index' indicates each feature dimension. Each plot shows the min-max range and the 5th-95th percentile range of the values within the same dimension.

where $S_X \in \mathbb{R}^{n \times 1}$ is the node-wise scale for $X$, $S_W \in \mathbb{R}^{d \times 1}$ is the feature-wise scale for $W$, and $\odot$ denotes the element-wise (Hadamard) product. Since $X$ is quantized node-wise and $W$ is quantized feature-wise, the quantized multiplication remains fast and efficient in *combination* phase.

**Challenge 1: Quantization along inner dimensions.** By contrast, node-wise quantization in the *aggregation* phase is more challenging. Existing GNN quantization (Zhu et al., 2022; Feng et al., 2020) methods instead apply column-wise quantization to the intermediate activation matrix $X_c$, because the columns correspond to the outer dimension in message passing (see Figure 4a). While this approach is computationally advantageous, it may fail to preserve the precision of activations. Specifically, applying node-wise quantization for the aggregation step,

$$\tilde{A} \cdot X_c \; \approx \; \mathrm{diag}(S_{\tilde{A}}) \cdot \tilde{A}^Q \; \cdot \; \mathrm{diag}(S_{X_c}) \cdot X_c^Q, \tag{5}$$

introduces the diagonal matrix $\mathrm{diag}(S_{X_c})$ within the multiplication. Unlike Equation 4, this cannot be computed using integer matrix multiplication units with common methods (Jacob et al., 2018). To capitalize on the precision benefits of node-wise quantization while also preserving computational efficiency, TopGQ proposes a novel method, *dual-axis scale absorption* (Section 5.2).

**Challenge 2: Generalization on unseen nodes.** For practical inductive settings (Section 2), contrary to transductive settings, GNN encounters nodes unseen at training time. To deal with unseen nodes, there can be two approaches to obtain accurate quantization parameters for each node:

*(i) On-the-Fly Quantization Parameter Computation.* A straightforward approach is to dynamically compute quantization parameters per node during inference. For each intermediate activation, every row of $X^{(l)}$ and $X_c^{(l)}$ is scanned, and the minimum and maximum values of each node are empirically determined to obtain scales and zero-points. While this ensures low quantization errors, it is less preferred as it causes runtime overhead that might counteract efficiency gains by quantization.

*(ii) Precomputed Mapping.* An alternative is to precompute a set of quantization parameters at calibration time and map each unseen node to one of its entries, typically to one from the training set nodes. Before inference, we can perform a simple lookup to retrieve and prepare the appropriate parameters for each activation. Nonetheless, this requires an accurate low-complexity *node index* $\phi(\cdot)$ such that nodes with similar index values exhibit similar feature statistics. TopGQ chooses this precomputed mapping approach, where we design a novel *Topology-Aware Pairwise Index (TopPIN)* that simply uses local topology for lightweight computation (Section 5.3). TopPIN ensures that unseen nodes are assigned adequate quantization parameters at low inference overhead.

## 5 TopGQ methodology

### 5.1 Overall framework of TopGQ

Figure 3 illustrates the overall process of TopGQ. In the calibration phase (Figure 3a), we first compute a topology-based value $\mathrm{TopPIN}(v)$ for each node $v$, which we define in Section 5.3. Based on these index values, we then calculate node-wise quantization parameters $(s_v, z_v)$ as described in Section 2. If multiple nodes share the same TopPIN, we aggregate the statistics by taking the global

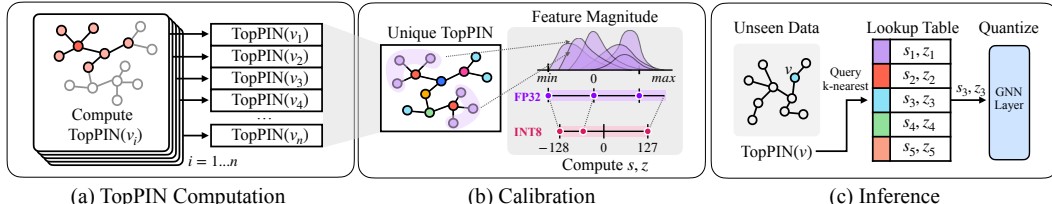

(a) TopPIN Computation  (b) Calibration  (c) Inference

Figure 3: The process of topology-aware quantization. (a) shows group generation using topological characteristics: TopPIN. Each color is used to denote each group. (b) shows the calibration process to achieve a set of quantization parameters for each group. (c) demonstrates how inference is done on unseen data by using the quantization parameters of the nearest groups with interpolation.

maximum and the global minimum (Figure 3b), ensuring the quantization parameters cover the full dynamic range. This gives us a pair of quantization parameters for each unique TopPIN value. Finally, during inference, we only need to compute the $\mathrm{TopPIN}(v)$ for each unseen node $v$ and use it as a key to retrieve the appropriate quantization parameters (Figure 3c). For this, we retrieve the $k$-nearest TopPIN groups and interpolate among their parameters. Such design is built on the idea that nodes with similar $\mathrm{TopPIN}(v)$ values exhibit similar activation distribution, which we theoretically demonstrate in Section 5.3.

On top of that, we apply dual-axis scale absorption which preserves the accuracy and efficiency during inference. Dual-axis scale absorption mimics the effect of node-wise quantization, while actually using feature-wise quantization to be compatible with integer matrix multiplication. This requires the computation of quantization scales along both axes, which are also calibrated via TopPIN. We demonstrate this process in detail in Section 5.2.

## 5.2 SELECTIVE DUAL-AXIS SCALE ABSORPTION

Node-wise quantization assigns quantization parameters per node, helping preserve diverse feature magnitudes. However, as seen in Equation (5), aggregation with naive node-wise quantization does not support integer operations. Given this, we aim to design *dual-axis scale absorption*, a technique that preserves both integer-operation speedups and node-wise quantization effects.

To account for the differing magnitude of node features (Figure 2), we employ a node-wise scale factor $S_N \in \mathbb{R}^{N \times 1}$, where $S_N$ consists of the maximum feature value for each node. Specifically, we scale $X_\mathrm{c}$ to $X_\mathrm{c}'$ with $S_N$, i.e., $X_\mathrm{c}' = \mathrm{diag}^{-1}(S_N) \cdot X_\mathrm{c}$. Then, to eliminate any obstacle terms preventing integer operations, $S_N$ is merged to the given static adjacency matrix, $\tilde{A} \in \mathbb{R}^{N \times N}$. The operation is as follows:

$$\tilde{A} \cdot X_\mathrm{c} = (\tilde{A} \cdot \mathrm{diag}(S_N)) \cdot X_\mathrm{c}' = \tilde{A}_{X_\mathrm{c}} \cdot X_\mathrm{c}' \quad (6)$$

After merging $S_N$ to $\tilde{A}$, we can conduct integer matrix multiplication for two matrices, $\tilde{A}_{X_\mathrm{c}}$ and $X_\mathrm{c}'$ with corresponding quantization parameters $S_{\tilde{A}_{X_\mathrm{c}}} \in \mathbb{R}^{N \times 1}$, and $S_{X_\mathrm{c}'} \in \mathbb{R}^{1 \times d}$:

$$\tilde{A}_{X_\mathrm{c}} \cdot X_\mathrm{c}' \quad (7)$$
$$\approx (\mathrm{diag}(S_{\tilde{A}_{X_\mathrm{c}}}) \cdot \tilde{A}_{X_\mathrm{c}}^Q) \cdot (X_\mathrm{c}'^Q \cdot diag(S_{X_\mathrm{c}'})) \quad (8)$$
$$= (S_{\tilde{A}_{X_\mathrm{c}}} \cdot S_{X_\mathrm{c}'}) \odot (\tilde{A}_{X_\mathrm{c}}^Q \cdot X_\mathrm{c}'^Q). \quad (9)$$

In the calibration process, TopGQ adaptively chooses between dual-axis and feature-wise quantization for $X_\mathrm{c}$ for each GNN layer. TopGQ evaluates both configurations by measuring the mean squared error

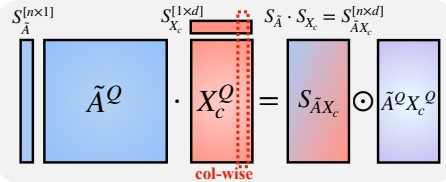

(a) Column-wise activation quantization

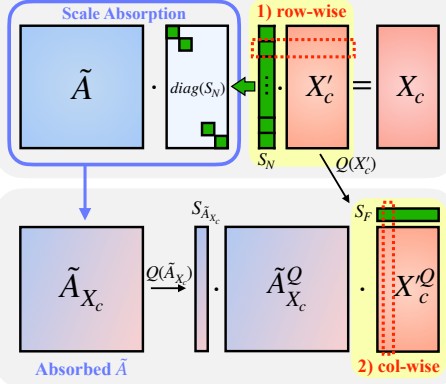

(b) Quantization via dual-axis scale absorption

Figure 4: Comparing quantization approaches at *aggregation* phase.

(MSE) between the original floating-point activations and their quantized counterparts. The configuration with lower MSE is saved for inference. When dual-axis scale absorption is selected, the scaling elements for $S_N$ are calibrated like the quantization parameters. When dual-axis scale absorption is used at inference, $X_c$ can be immediately quantized with $S_N \cdot S_{X_c'} \in \mathbb{R}^{N \times d}$, which acts like an element-wise quantization parameters for $X_c$.

## 5.3 TopPIN: A LIGHTWEIGHT INDEX FOR UNSEEN NODES

To support quantization in inductive settings, we devise TopPIN, a lightweight index that maps unseen nodes to existing train set nodes used at calibration. The formulation of TopPIN is as follows:

$$\text{TopPIN}(v) = \Big( d(v), \ \frac{1}{d(v)} \sum_{v_k \in \mathcal{N}(v)} \frac{1}{d(v_k)} \Big). \tag{10}$$

Its formulation is derived from two aggregation cases, where connections from neighbor nodes affect the target node's feature distribution. We present this link of local structure and node features at Theorem 1. We can leverage the link to a node-level index function $\phi : V \to \mathbb{R}$, that captures the expected per-node feature variance from local topology. The proof is provided in Section A.

**Theorem 1** (Node index $\phi$ for GNN activation). *Let $G = (V, E)$ be an undirected graph. For $\tilde{A}$, we separately consider unnormalized and normalized cases. Consider a GNN of the form*

$$X^{(l+1)} \ = \ ReLU\big( \tilde{A} \, X^{(l)} \, W^{(l)} \big),$$

*where the hidden dimensions $d_l$ are sufficiently large $(d_l \gg 1)$ for all layers $l$, and each entry of $X^{(0)}$ and $W^{(l)}$ is drawn independently from a distribution with zero mean and finite variance. Then, for each layer $l$ of the GNN, define the scalar function $\phi : V \to \mathbb{R}$ by:*

$$\phi(v) = \sum_{v_{k_1} \in \mathcal{N}(v)} w(v, v_{k_1}) \sum_{v_{k_2} \in \mathcal{N}(v_{k_1})} w(v_{k_1}, v_{k_2}) \cdots \sum_{v_{k_l} \in \mathcal{N}(v_{k_{l-1}})} w(v_{k_{l-1}}, v_{k_l}),$$

$$with \quad w(v, u) = \begin{cases} w_1(v, u) = 1, & \tilde{A} = A + I_n, \\ w_2(v, u) = \dfrac{1}{d(v) \, d(u)}, & \tilde{A} = D^{-\frac{1}{2}}(A + I_n)D^{-\frac{1}{2}} \end{cases}$$

*Asymptotically, the probability distribution of each row of $X^{(l)}W^{(l)}$ and $\tilde{A}X^{(l)}W^{(l)}$ is determined solely by $\phi(v)$.*

$\phi(\cdot)$ implies that similar quantization parameters can be made when $\phi$-values align among nodes. We show this correlation in Theorem 2.

**Theorem 2** (Node index $\phi$ and GNN quantization parameters). *The expected per-node quantization parameters for $X^{(l)}$ and $X^{(l)} W^{(l)}$ vary as uniformly continuous functions of $\phi(\cdot)$. In particular:*

- *If $\phi(u) = \phi(v)$, then $\mathbb{E}[s_u] = \mathbb{E}[s_v]$ and $\mathbb{E}[z_u] = \mathbb{E}[z_v]$, where $(s_u, z_u)$ and $(s_v, z_v)$ are the respective scale and zero-point parameters of nodes $u$ and $v$.*

- *More generally, if $|\phi(u) - \phi(v)| < \delta$, then $|\mathbb{E}[s_u] - \mathbb{E}[s_v]| < \epsilon$ and $|\mathbb{E}[z_u] - \mathbb{E}[z_v]| < \epsilon$, for any desired $\epsilon > 0$, by uniform continuity.*

Hence, Nodes with similar $\phi$-values have similar calibrated quantization parameters in expectation. The proof can be found in Section B.

While $\phi$ faithfully reflects the expected per-node quantization statistics, its direct usage is computationally expensive for multi-layer GNN inference. Therefore, we derive TopPIN from $\phi$, which is notably fast to compute as it is composed with the first summation terms from Theorem 1. Consequently, TopPIN is also noted as follows. Refer to Section C for the details:

$$\text{TopPIN}(v) = \Big( \sum_{u \in \mathcal{N}(v)} w_1(v, u), \ \sum_{u \in \mathcal{N}(v)} w_2(v, u) \Big). \tag{11}$$

Table 1: Comparison of quantization accuracy and time for node classification task

| | Cora | | | | | | Citeseer | | | | | | PubMed | | | | | |
|---|---|---|---|---|---|---|---|---|---|---|---|---|---|---|---|---|---|---|
| | GCN | | GAT | | GIN | | GCN | | GAT | | GIN | | GCN | | GAT | | GIN | |
| | Acc. | Q.T. | Acc. | Q.T. | Acc. | Q.T. | Acc. | Q.T. | Acc. | Q.T. | Acc. | Q.T. | Acc. | Q.T. | Acc. | Q.T. | Acc. | Q.T. |
| FP32 | 80.14 | - | 80.36 | - | 79.76 | - | 76.46 | - | 76.42 | - | 77.06 | - | 87.12 | - | 87.66 | - | 88.93 | - |
| **INT8** | | | | | | | | | | | | | | | | | | |
| SGQ | 79.93 | (6.4s) | 80.30 | (7.7s) | 78.35 | (5.4s) | 76.21 | (9.3s) | 76.20 | (9.6s) | 76.04 | (11.3s) | 86.37 | (14.8s) | **88.96** | (19.6s) | **88.89** | (16.6s) |
| DQ | 78.94 | (11.5s) | 78.66 | (14.6s) | 67.56 | (18.5s) | 75.62 | (25.7s) | 23.66 | (28.4s) | 65.64 | (50.6s) | **88.26** | (50.6s) | 88.11 | (60.6s) | 88.32 | (188.9s) |
| EPQ | 77.52 | (20.6s) | 75.60 | (23.0s) | 56.78 | (20.8s) | 75.26 | (72.4s) | 75.30 | (73.2s) | 56.68 | (72.9s) | 83.16 | (106.5s) | 82.54 | (107.7s) | 77.26 | (108.7s) |
| $A^2Q$ | 79.66 | (3.5s) | 75.29 | (6.3s) | 69.16 | (3.3s) | 75.72 | (3.5s) | 74.34 | (7.1s) | 67.09 | (4.0s) | 85.20 | (8.0s) | 83.38 | (8.2s) | 88.36 | (13.1s) |
| QLR | 79.12 | (4.0s) | 78.84 | (6.4s) | 60.60 | (3.3s) | 40.66 | (3.8s) | 43.76 | (6.2s) | 56.92 | (3.3s) | 83.14 | (4.1s) | 86.66 | (6.6s) | 86.84 | (6.8s) |
| DRA | 79.76 | (1.7s) | 80.20 | (3.6s) | 78.83 | (1.7s) | 76.26 | (1.6s) | **76.31** | (3.1s) | 76.55 | (1.7s) | 86.75 | (1.7s) | 88.69 | (3.3s) | 87.73 | (1.7s) |
| TopGQ | **79.96** | (0.2s) | **80.63** | (0.2s) | **80.01** | (0.2s) | **76.52** | (0.2s) | **76.31** | (0.2s) | **77.08** | (0.2s) | 87.14 | (0.2s) | 87.42 | (0.2s) | 88.82 | (0.3s) |
| **INT4** | | | | | | | | | | | | | | | | | | |
| SGQ | 78.73 | (6.4s) | 77.92 | (6.8s) | 76.93 | (5.4s) | **76.31** | (10.4s) | 75.54 | (13.3s) | 75.47 | (11.3s) | 85.12 | (14.8s) | 85.14 | (19.6s) | **88.33** | (16.6s) |
| DQ | 78.54 | (11.5s) | 77.90 | (14.5s) | 65.98 | (18.5s) | 23.54 | (25.5s) | 23.58 | (28.6s) | 46.02 | (49.5s) | **87.78** | (50.2s) | **87.41** | (60.3s) | 86.94 | (190.6s) |
| EPQ | 76.32 | (20.6s) | 74.52 | (23.1s) | 32.98 | (20.7s) | 74.92 | (72.3s) | 74.70 | (73.2s) | 46.66 | (72.9s) | 81.26 | (106.4s) | 81.36 | (107.5s) | 41.18 | (108.8s) |
| $A^2Q$ | 50.00 | (3.6s) | 45.64 | (6.3s) | 68.76 | (3.2s) | 43.52 | (3.5s) | 58.50 | (7.1s) | 62.50 | (4.0s) | 70.08 | (4.0s) | 72.52 | (8.4s) | 84.74 | (8.2s) |
| QLR | 76.64 | (3.7s) | 78.76 | (6.2s) | 68.38 | (3.4s) | 37.40 | (3.9s) | 40.06 | (6.1s) | 62.90 | (3.3s) | 74.08 | (3.3s) | 86.26 | (6.4s) | 76.90 | (6.9s) |
| DRA | 77.02 | (1.7s) | 74.35 | (3.3s) | 57.29 | (1.7s) | 74.10 | (1.6s) | 72.92 | (3.1s) | 61.10 | (1.7s) | 75.09 | (1.7s) | 72.44 | (3.4s) | 36.90 | (1.7s) |
| TopGQ | **78.84** | (0.2s) | **78.56** | (0.2s) | **79.34** | (0.2s) | 75.96 | (0.2s) | **76.24** | (0.2s) | **76.92** | (0.2s) | 86.92 | (0.2s) | 86.91 | (0.2s) | 87.72 | (0.3s) |

∗Q.T.: Quantization Time, SGQ: SGQuant, DQ: Degree-Quant, EPQ: EPQuant

# 6 EXPERIMENTAL RESULTS

## 6.1 EXPERIMENTAL SETTINGS

We evaluate TopGQ on both node-level and graph-level classification tasks, and compare it with five QAT baselines: SGQuant (Feng et al., 2020), Degree-Quant (Tailor et al., 2020), EPQuant (Huang et al., 2022), $A^2Q$ (Zhu et al., 2022), and QLR (Wang et al., 2023), and one recent PTQ baseline: DRA (Jeddi et al., 2024). For node classification, we use the Cora, CiteSeer, PubMed, Reddit, ogbn-products, and MAG240M datasets; and for graph classification, we use IMDB-BINARY, and COLLAB datasets. For large-scale and hyper-scale datasets, we evaluate GNN architectures (e.g., GCN, GraphSAGE, R-GAT) that were introduced as baseline architectures in the original ogb-benchmark papers (Hu et al., 2020; 2021). For other datasets, we calibrate a fully-trained GCN (Kipf & Welling, 2016), GAT (Veličković et al., 2018), GIN (Xu et al., 2019), and GraphSAGE (Hamilton et al., 2017) for 4-bit and 8-bit integer quantization; the bitwidth is fixed across all layers for fair comparison. Tasks with datasets except MAG240M were conducted in the inductive setting, which reflects a more practical use of quantization. Node classification with MAG240M was evaluated with the original transductive setting introduced in the ogb-lsc (Hu et al., 2021) challenge. Further experimental results and details are provided in the Appendix.

## 6.2 EVALUATION RESULTS OF NODE-LEVEL TASKS

Table 1 reports results on the conventional node-level datasets commonly used in baselines. Even though the datasets are relatively small, TopGQ is the fastest, taking less than a second for quantization while achieving comparable or superior accuracy compared to the baselines. For larger datasets (Table 3), the difference is clearer, where baselines take up to hours (3.27h, Reddit, GS) for quantization whilst TopGQ takes less than a minute. For INT4, TopGQ exceeds all baselines in terms of both accuracy and speed. Notably, while all methods suffer from low accuracy in ogbn-products due to the small number of train nodes, TopGQ shows best accuracy due to its ability to adapt to unseen nodes. For brevity, we report the full experiment set in the Appendix.

Table 2: Quantized validation accuracy and time on MAG240M node-classification task with R-GAT

| Method | Type | Acc. | Q. Time |
|---|---|---|---|
| FP32 | — | 69.66 | – |
| SGQ | QAT | 46.76 | 5.50 **days** |
| $A^2Q$ | QAT | 57.97 | 2.46 **days** |
| QLR | QAT | 68.10 | 1.97 **days** |
| DRA | PTQ | 66.13 | 2.06 **days** |
| TopGQ | PTQ | **69.14** | 0.98 **hours** |

Table 3: Comparison of node classification task with large-scale graphs

| Method | INT8 | | | | | | | | INT4 | | | | | | | |
|---|---|---|---|---|---|---|---|---|---|---|---|---|---|---|---|---|
| | Reddit, GCN | | Reddit, GS | | ogbn, GCN | | ogbn, GS | | Reddit, GCN | | Reddit, GS | | ogbn, GCN | | ogbn, GS | |
| | Acc. | Q.Time | Acc. | Q.Time | Acc. | Q.Time | Acc. | Q.Time | Acc. | Q.Time | Acc. | Q.Time | Acc. | Q.Time | Acc. | Q.Time |
| FP32 | 94.40 | – | 95.09 | – | 71.25 | – | 70.33 | – | 94.40 | – | 95.09 | – | 71.25 | – | 70.33 | – |
| SGQ | 92.10 | (4.64m) | 92.01 | (3.27h) | 39.13 | (2.52m) | 58.80 | (2.11h) | 43.00 | (4.84m) | 87.42 | (3.27h) | 6.14 | (2.57m) | 27.95 | (2.13h) |
| DQ | 87.01 | (10.59m) | 90.53 | (16.35h) | **72.34** | (14.05m) | 70.17 | (6.55h) | 64.18 | (10.55m) | 89.61 | (16.33h) | 36.66 | (13.93m) | **69.90** | (6.52h) |
| EPQ | 80.29 | (5.30m) | 93.11 | (1.36h) | 49.33 | (53.68s) | 56.83 | (52.42m) | 22.02 | (5.29m) | 79.61 | (1.36h) | 26.96 | (53.58s) | 26.97 | (52.40m) |
| $A^2Q$ | 73.71 | (4.12m) | 75.13 | (2.83h) | 50.78 | (83.94s) | 60.15 | (1.67h) | 23.24 | (4.12m) | 67.94 | (2.83h) | 25.95 | (83.30s) | 31.32 | (1.66h) |
| QLR | 94.21 | (72.86s) | **95.11** | (31.82m) | 66.48 | (76.30s) | 63.85 | (50.15m) | 86.95 | (72.79s) | 81.68 | (31.78m) | 27.36 | (76.12s) | 29.38 | (50.16m) |
| DRA | 93.15 | (42.99s) | 94.36 | (23.71m) | 36.22 | (41.63s) | 47.70 | (44.97m) | 1.75 | (42.82s) | 5.31 | (23.71m) | 3.12 | (41.61s) | 26.40 | (44.96m) |
| TopGQ | **94.41** | (1.88s) | 94.55 | (35.79s) | 71.33 | (1.16s) | **70.31** | (34.88s) | **93.05** | (1.87s) | **89.88** | (35.28s) | **39.03** | (1.16s) | 61.83 | (34.90s) |

∗Q.T.: Quantization Time, SGQ: SGQuant, DQ: Degree-Quant, EPQ: EPQuant

Table 4: Comparison of quantization accuracy and time for the graph-classification benchmarks

| | Method | IMDB-BINARY | | | | | | COLLAB | | | | | |
|---|---|---|---|---|---|---|---|---|---|---|---|---|---|
| | | GCN | | GAT | | GIN | | GCN | | GAT | | GIN | |
| | | Acc. | Q.Time | Acc. | Q.Time | Acc. | Q.Time | Acc. | Q.Time | Acc. | Q.Time | Acc. | Q.Time |
| FP32 | - | 79.58 | – | 77.36 | – | 79.72 | – | 82.54 | – | 79.99 | – | 82.31 | – |
| INT8 | SGQ | 68.28 | (5.88m) | 68.60 | (14.65m) | 68.26 | (6.74m) | 80.96 | (39.35m) | 80.10 | (2.12h) | 81.80 | (40.26m) |
| | DQ | 77.32 | (8.98m) | 75.12 | (25.53m) | 76.00 | (9.08m) | 82.30 | (2.39h) | 80.30 | (8.14h) | 81.62 | (2.30h) |
| | EPQ | 76.30 | (13.99m) | 74.00 | (18.87m) | 76.70 | (14.17m) | 82.36 | (2.99h) | 79.81 | (0.81h) | 77.66 | (2.95h) |
| | $A^2Q$ | 75.12 | (3.24m) | 74.91 | (8.18m) | 75.97 | (3.78m) | 64.10 | (14.75m) | 74.35 | (2.03h) | 80.21 | (14.50m) |
| | QLR | 75.50 | (3.44m) | 74.40 | (7.30m) | 74.50 | (3.70m) | 81.98 | (13.04m) | 75.47 | (0.81h) | 81.70 | (9.42m) |
| | DRA | 78.88 | (2.24m) | **77.70** | (4.38m) | 78.52 | (2.29m) | 82.08 | (11.49m) | **80.78** | (0.50h) | 82.18 | (10.19m) |
| | TopGQ | **79.34** | (2.18s) | 76.58 | (5.58s) | **79.50** | (2.05s) | **82.52** | (13.86s) | 77.46 | (47.32s) | **82.28** | (11.71s) |
| INT4 | SGQ | 67.64 | (5.89m) | 67.49 | (14.71m) | 63.72 | (6.71m) | 78.14 | (38.87m) | 78.22 | (2.11h) | 72.06 | (40.44m) |
| | DQ | 76.02 | (9.03m) | 74.71 | (26.07m) | **75.98** | (9.22m) | 73.24 | (2.40h) | **79.51** | (8.16h) | 77.61 | (2.31h) |
| | EPQ | 74.80 | (13.95m) | 74.10 | (18.80m) | 64.80 | (14.12m) | 65.54 | (2.98h) | 71.63 | (0.82h) | 65.94 | (2.94h) |
| | $A^2Q$ | 74.09 | (3.13m) | 72.80 | (8.11m) | 75.62 | (3.79m) | 69.32 | (14.94m) | 74.96 | (2.02h) | 74.78 | (14.40m) |
| | QLR | 73.40 | (3.46m) | 74.00 | (7.26m) | 73.50 | (3.74m) | **81.92** | (13.09m) | 72.87 | (0.80h) | **79.32** | (9.47m) |
| | DRA | 74.32 | (2.22m) | 75.58 | (4.34m) | 70.28 | (2.30m) | 64.16 | (11.45m) | 78.75 | (0.50h) | 66.24 | (10.18m) |
| | TopGQ | **76.71** | (2.08s) | **75.72** | (5.69s) | 76.00 | (2.13s) | 81.75 | (13.85s) | 73.33 | (47.29s) | 77.39 | (11.71s) |

∗Q.T.: Quantization Time, SGQ: SGQuant, DQ: Degree-Quant, EPQ: EPQuant

We further emphasize the benefit TopGQ by using a hyper-scale graph with 240 million nodes in Table 2. Existing methods take at least 1.97 days, up to 5.50 days to quantize a GNN on such a hyper-scale graph, while TopGQ cuts it down to 0.98 hour, showing at least 49× speedup. At the same time, TopGQ presents a negligible difference to the FP32 model.

## 6.3 EVALUATION RESULTS OF GRAPH-LEVEL TASKS

Table 4 presents the graph-level classification results on IMDB-BINARY and COLLAB. TopGQ significantly improves quantization speed while maintaining competitive task performance. For instance, while EPQuant is the strongest baseline in GCN COLLAB, it takes 2.39 hours for quantization. However, TopGQ shows superior accuracy while cutting down the overhead to 13.86 seconds. While TopGQ takes the least time to quantize, in many cases TopGQ also shows the best accuracy with minimal degradation compared to FP32. We attribute this to TopGQ's explicit integration of GNN-specific considerations – leveraging TopPIN to effectively capture local topological structures, while QAT baselines largely neglect these properties. Overall, the experimental results for node and graph classification tasks demonstrate that TopGQ provides a robust balance between accuracy and quantization speed, making it well-suited for both small and large-scale GNN tasks.

## 6.4 EVALUATION RESULTS OF INFERENCE LATENCY

Table 5 report the inference latency of TopGQ and competing baselines with the minibatch setting of large-scale graphs. Measurements were conducted on both GPU and edge devices, reflecting practical scenarios for quantized GNN deployment. A key observation is that $A^2Q$ and on-the-fly PTQ are expensive and slow. This is because both methods require row-wise scans of intermediate activations

Table 5: GCN inference time (sec) on GPU (RTX4090) and edge device (Jetson AGX Orin)

| Method | Type | GPU (RTX 4090) | | | | Edge (Jetson AGX Orin) | | | |
|---|---|---|---|---|---|---|---|---|---|
| | | Reddit | | ogbn-products | | Reddit | | ogbn-products | |
| | | Time | Speedup | Time | Speedup | Time | Speedup | Time | Speedup |
| FP32 | – | 2.18 | – | 34.51 | – | 51.68 | – | 754.09 | – |
| Degree-Quant | QAT | 1.41 | 1.55× | 20.37 | 1.69× | 33.65 | 1.54× | 463.96 | 1.63× |
| $A^2Q$ | QAT | 1.96 | 1.11× | 27.74 | 1.24× | 48.42 | 1.07× | 635.15 | 1.19× |
| SGQuant | QAT | 1.42 | 1.54× | 20.53 | 1.68× | 34.17 | 1.51× | 470.89 | 1.60× |
| On-the-fly PTQ | PTQ | 2.04 | 1.07× | 27.73 | 1.24× | 54.88 | 0.94× | 689.82 | 1.09× |
| TopGQ | PTQ | 1.42 | 1.54× | 20.53 | 1.68× | 34.23 | 1.51× | 473.25 | 1.59× |

Table 6: Accuracy and index computation time comparison on IMDB-BINARY

| Bit | Node Index | GCN | GAT | GIN | Time |
|---|---|---|---|---|---|
| INT4 | Naive PTQ | 60.14 | 51.72 | 56.50 | - |
| | Betweenness | 50.00 | 71.42 | 50.00 | 1.85s |
| | Closeness | 72.90 | 70.62 | 67.78 | 1.48s |
| | Katz | 69.34 | 57.48 | 72.58 | 20.04s |
| | TopPIN | **76.71** | **75.72** | **76.00** | **0.00059s** |

Table 7: Ablation Study of TopGQ

| Bit | Method | Reddit | | ogbn-products | |
|---|---|---|---|---|---|
| | | GCN | GS | GCN | GS |
| INT4 | Naive PTQ | 3.79 | 2.97 | 1.33 | 24.74 |
| | Only TopPIN | 93.05 | 85.83 | 1.43 | 52.45 |
| | TopGQ | **93.05** | **89.88** | **39.03** | **63.18** |

to derive quantization parameters. These findings highlight the importance of storing quantization parameters and retrieving them via efficient mapping function. TopGQ leverages TopPIN, where index computation incurs negligible overhead, enabling efficient inference (Section 5.3).

## 6.5 ANALYSIS ON TOPPIN AND ABLATION STUDY

We assess the effectiveness of TopPIN by comparing it against a naive PTQ strategy as well as commonly used centrality measures, including betweenness, closeness, and Katz centrality (Table 6). We report both the accuracy and the total time to compute values for all nodes in the dataset. The naive PTQ approach, which relies on a single global quantization parameter, shows significant accuracy degradation due to high variance in node magnitudes. The conventional centrality measures may mitigate the accuracy degradation, compared to naive PTQ. However, they require costly global graph traversal for each node, making them impractical for quantized inference. In contrast, TopPIN only depends on 1-hop neighborhood information, thereby significantly reducing the computational overhead. Despite its lightweight design, TopPIN outperforms other centrality baselines, highlighting its practicality and effectiveness as an indexing strategy for GNN quantization.

Finally, Table 7 shows the ablation study, where each row corresponds to the incremental addition of TopPIN and selective dual-axis scale absorption to the naive PTQ baseline, ultimately forming TopGQ. While naive PTQ fails to effectively exploit the quantization bins under the INT4 configuration, TopPIN mitigates this limitation by leveraging topology to better preserve node features. However, as graph size increases (e.g., ogbn-products with GCN), TopPIN alone proves insufficient. By further introducing dual-axis scale absorption, the node-wise quantization effects are consistently preserved across GNN layers, leading to additional accuracy recovery.

## 7 CONCLUSION

In this work, we introduce TopGQ, a topology-aware post-training quantization framework for GNNs that eliminates the need for retraining while preserving high accuracy. By leveraging a novel node index (TopPIN) and the dual-axis scale absorption mechanism, TopGQ can handle unseen node features of differing magnitudes. This node-level strategy enables fast and precise activation quantization while preserving the computational benefits of integer operations. Extensive experiments across various GNN architectures and datasets show that TopGQ achieves quantization-aware training (QAT)-level accuracy, while reducing quantization time by an order of magnitude compared to prior works. These results establish TopGQ as a practical and scalable solution for efficient GNN inference, including large-scale and hyper-scale graph datasets.

## 8 REPRODUCIBILITY STATEMENT

The derivation of TopPIN comes from the theoretically grounded link between local structure and expected node feature variance. We provide the related theorems and corresponding proof in the Appendix. The code to reproduce the results of TopGQ with citation datasets can be downloaded at `https://anonymous.4open.science/r/topgq-code-3CF1`. We provide a README file for the environment setups, TopPIN generation, and commands for experiments with TopGQ. Further details for the experiments are at Section I.

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

## A   THE PROOF OF THEOREM 1

**Theorem** (Node index $\phi$ for GNN activation). *Let $G = (V, E)$ be an undirected graph. For $\tilde{A}$, we separately consider unnormalized and normalized cases. Consider a GNN of the form*

$$X^{(l+1)} \;=\; ReLU\big( \tilde{A}\, X^{(l)}\, W^{(l)} \big),$$

*where the hidden dimensions $d_l$ are sufficiently large ($d_l \gg 1$) for all layers $l$, and each entry of $X^{(0)}$ and $W^{(l)}$ is drawn independently from a distribution with zero mean and finite variance. Then, for each layer $l$ of the GNN, define the scalar function $\phi : V \to \mathbb{R}$ by:*

$$\phi(v) = \sum_{v_{k_1} \in \mathcal{N}(v)} w(v, v_{k_1}) \sum_{v_{k_2} \in \mathcal{N}(v_{k_1})} w(v_{k_1}, v_{k_2}) \cdots \sum_{v_{k_l} \in \mathcal{N}(v_{k_{l-1}})} w(v_{k_{l-1}}, v_{k_l}),$$

$$with \quad w(v, u) = \begin{cases} w_1(v, u) = 1, & \tilde{A} = A + I_n, \\ w_2(v, u) = \dfrac{1}{d(v)\, d(u)}, & \tilde{A} = D^{-\frac{1}{2}}(A + I_n)D^{-\frac{1}{2}} \end{cases}$$

*Asymptotically, the probability distribution of each row of $X^{(l)}W^{(l)}$ and $\tilde{A}X^{(l)}W^{(l)}$ is determined solely by $\phi(v)$.*

*Proof.*  **Case 1:** If $\tilde{A} = A + I_n$ (**unnormalized**), then

$$\phi(v) \;=\; \sum_{v_{k_1} \in \mathcal{N}(v)} \sum_{v_{k_2} \in \mathcal{N}(v_{k_1})} \cdots \sum_{v_{k_l} \in \mathcal{N}(v_{k_{l-1}})} 1.$$

**Case 2:** If $\tilde{A} = D^{-\frac{1}{2}}(A + I_n)\, D^{-\frac{1}{2}}$ (**normalized**), then

$$\phi(v) = \frac{1}{d(v)} \sum_{k_1 \in \mathcal{N}(v)} \left[ \frac{1}{d(v_{k_1})^2} \sum_{k_2 \in \mathcal{N}(v_{k_1})} \left( \frac{1}{d(v_{k_2})^2} \quad \cdots \quad \sum_{k_l \in \mathcal{N}(v_{k_{l-1}})} \frac{1}{d(v_{k_l})} \right) \right].$$

We consider the $L$-layer GNN

$$X^{(l+1)} \;=\; ReLU\big( \tilde{A}\, X^{(l)}\, W^{(l)} \big), \quad l = 0, \ldots, L-1,$$

$$\tilde{A} \;\in\; \left\{ A + I_n, \;\; D^{-\frac{1}{2}}(A + I_n)D^{-\frac{1}{2}} \right\}.$$

Assume that each entry of the initial feature matrix $X^{(0)} \in \mathbb{R}^{n \times d_0}$ is drawn i.i.d. from a zero-mean distribution $\mathcal{D}_x(0, \sigma_x^2)$, and each entry of the weight matrices $W^{(l)} \in \mathbb{R}^{d_l \times d_{l+1}}$ is drawn i.i.d. from a zero-mean distribution $\mathcal{D}_l(0, \sigma_l^2)$. Denote

$$Z^{(l)} \;=\; X^{(l)} W^{(l)}, \quad Y^{(l)} \;=\; \tilde{A}\, Z^{(l)}, \quad X^{(l+1)} \;=\; ReLU\big( Y^{(l)} \big).$$

for notational simplicity. We prove in several steps that each row of $Z^{(l)}$ and $Y^{(l)}$ is (approximately) a zero-mean Gaussian whose variance depends only on a node-specific function $\phi(\cdot)$, and that this leads to the stated continuity property of expected quantization parameters.

**Step 1: Propagation for 1st layer of GNN.**   Consider $Z^{(0)} \;=\; X^{(0)} W^{(0)}$. For each entry,

$$Z_{ij}^{(0)} \;=\; \sum_{\alpha=1}^{d_0} X_{i\alpha}^{(0)} W_{\alpha j}^{(0)}.$$

Since $X_{i\alpha}^{(0)}$ and $W_{\alpha j}^{(0)}$ are i.i.d. with zero mean and finite variance, it follows that

$$\mathbb{E}[X_{i\alpha}^{(0)} W_{\alpha j}^{(0)}] = 0, \quad \mathrm{Var}(X_{i\alpha}^{(0)} W_{\alpha j}^{(0)}) = \mathbb{E}[(X_{i\alpha}^{(0)})^2]\mathbb{E}[(W_{\alpha j}^{(0)})^2] = \sigma_x^2 \sigma_0^2$$

By the Central Limit Theorem (CLT), $Z_{ij}^{(0)}$ is approximately $\mathcal{N}(0, \; d_0 \sigma_x^2 \sigma_0^2)$.

Next, we consider the multiplication by the augmented adjacency matrix. For instance, $\tilde{A} = D^{-\frac{1}{2}}(I + A)D^{-\frac{1}{2}}$.

$$Y^{(0)} = \tilde{A} Z^{(0)}, \quad Y^{(0)}_{ij} = \sum_{\alpha=1}^{n} \tilde{A}_{i\alpha} Z^{(0)}_{\alpha j} = \sum_{\{k|v_k \in \mathcal{N}(v_i)\}} \frac{1}{\sqrt{d(v_i)\, d(v_k)}} Z^{(0)}_{kj}.$$

Summing independent Gaussians preserves Gaussianity and yields

$$Y^{(0)}_{ij} \sim \mathcal{N}(0, [d_0 \sigma_x^2 \sigma_0^2] \frac{1}{d(v_i)} \sum_{v_k \in \mathcal{N}(v_i)} \frac{1}{d(v_k)}).$$

**Step 2: Subsequent layers and ReLU.** Next we consider for the second layer of GNN. Focusing on a particular entry $Z^{(1)}_{ij}$, we can write

$$Z^{(1)}_{ij} = \sum_{\alpha=1}^{d_1} X^{(1)}_{i\alpha} W^{(1)}_{\alpha j} = \sum_{\alpha=1}^{d_1} ReLU(Y^{(0)}_{i\alpha})\, W^{(1)}_{\alpha j}.$$

Since $W^{(1)}_{\alpha j} \sim \mathcal{N}(0, \sigma_1^2)$, we have :

$$\mathbb{E}\big[X^{(1)}_{i\alpha} W^{(1)}_{\alpha j}\big] = 0, \quad {}^{1}\mathrm{Var}\big(X^{(1)}_{i\alpha} W^{(1)}_{\alpha j}\big) = \mathbb{E}[(ReLU(Y^{(0)}_{i\alpha}))^2]\, \sigma_1^2 = \frac{[d_0 \sigma_x^2 \sigma_0^2] \frac{1}{d(v_i)} \sum_{v_k \in \mathcal{N}(v_i)} \frac{1}{d(v_k)}}{2} \sigma_1^2.$$

Summing over $\alpha = 1, \ldots, d_1$ and again invoking the CLT, we arrive at

$$Z^{(1)}_{ij} \sim \mathcal{N}(0, [\frac{d_0 d_1 \sigma_x^2 \sigma_0^2 \sigma_1^2}{2}] \frac{1}{d(v_i)} \sum_{v_k \in \mathcal{N}(v_i)} \frac{1}{d(v_k)}).$$

We then multiply by the adjacency matrix $\tilde{A}$ as before,

$$Y^{(1)} = \tilde{A} Z^{(1)}, \quad Y^{(1)}_{ij} = \sum_{\alpha=1}^{n} \tilde{A}_{i\alpha} Z^{(1)}_{\alpha j} = \sum_{\{k_1|v_{k_1} \in \mathcal{N}(v_i)\}} \frac{1}{\sqrt{d(v_i)\, d(v_{k_1})}} Z^{(1)}_{k_1 j}.$$

Thus for $Y^{(1)}_{ij}$ we have

$$Y^{(1)}_{ij} \sim \mathcal{N}(0, [\frac{d_0 d_1 \sigma_x^2 \sigma_0^2 \sigma_1^2}{2}] \frac{1}{d(v_i)} \sum_{v_{k_1} \in \mathcal{N}(v_i)} \frac{1}{d(v_{k_1})^2} \sum_{v_{k_2} \in \mathcal{N}(v_{k_1})} \frac{1}{d(v_{k_2})}).$$

One can easily see that through repetition we have

$$\phi(v_i) = \frac{1}{d(v_i)} \sum_{k_1 \in \mathcal{N}(v_i)} \left[ \frac{1}{d(v_{k_1})^2} \sum_{k_2 \in \mathcal{N}(v_{k_1})} \left( \frac{1}{d(v_{k_2})^2} \cdots \sum_{k_l \in \mathcal{N}(v_{k_{l-1}})} \frac{1}{d(v_{k_l})} \right) \right]$$

$$Y^{(l-1)}_{ij} \sim \mathcal{N}\left( 0, \frac{\left(\prod_{m=0}^{l-1} d_m\right) \sigma_x^2 \left(\prod_{m=0}^{l-1} \sigma_m^2\right)}{2^{l-1}} \cdot \phi(v_i) \right)$$

$$Z^{(l)}_{ij} \sim \mathcal{N}\left( 0, \frac{\left(\prod_{m=0}^{l} d_m\right) \sigma_x^2 \left(\prod_{m=0}^{l} \sigma_m^2\right)}{2^{l}} \cdot \phi(v_i) \right)$$

that is the distribution for each row of $Z^{(l)}$ and $Y^{(l)}$ are zero-mean Gaussian whose variance depends on the corresponding node-specific scalar function which is a constant factor of $\phi$.

---

[1]For a ReLU applied zero mean gaussian random variable $Y \sim \mathcal{N}(0, \sigma_y^2)$,

$$\mathbb{E}[(ReLU(Y))^2] = \int_0^{\infty} y^2\, \frac{1}{\sigma_y \sqrt{2\pi}} e^{-y^2/(2\sigma_y^2)}\, dy = \frac{\sigma_y^2}{2}$$

**Step 3 : Unnormalized adjacency.** For the case $\tilde{A} = A + I_n$, since each neighbor contribution is no longer scaled, the entire degree-based weighting disappears. That is,

$$\phi(v_i) = \sum_{v_{k_1} \in \mathcal{N}(v_i)} \sum_{v_{k_2} \in \mathcal{N}(v_{k_1})} \cdots \sum_{v_{k_l} \in \mathcal{N}(v_{k_{l-1}})} 1.$$

$$Y_{ij}^{(l-1)} \sim N\left(0, \frac{\left(\prod_{m=0}^{l-1} d_m\right) \sigma_x^2 \left(\prod_{m=0}^{l-1} \sigma_m^2\right)}{2^{l-1}} \cdot \phi(v_i)\right)$$

$$Z_{ij}^{(l)} \sim N\left(0, \frac{\left(\prod_{m=0}^{l} d_m\right) \sigma_x^2 \left(\prod_{m=0}^{l} \sigma_m^2\right)}{2^l} \cdot \phi(v_i)\right)$$

holds. $\qquad\square$

## B   THE PROOF OF THE THEOREM 2

**Theorem** (Node index $\phi$ and GNN quantization parameters). *The* expected *per-node quantization parameters for $X^{(l)}$ and $X^{(l)} W^{(l)}$ vary as uniformly continuous functions of $\phi(\cdot)$. In particular:*

- *If $\phi(u) = \phi(v)$, then $\mathbb{E}[s_u] = \mathbb{E}[s_v]$ and $\mathbb{E}[z_u] = \mathbb{E}[z_v]$, where $(s_u, z_u)$ and $(s_v, z_v)$ are the respective scale and zero-point parameters of nodes $u$ and $v$.*

- *More generally, if $|\phi(u) - \phi(v)| < \delta$, then $|\mathbb{E}[s_u] - \mathbb{E}[s_v]| < \epsilon$ and $|\mathbb{E}[z_u] - \mathbb{E}[z_v]| < \epsilon$, for any desired $\epsilon > 0$, by uniform continuity.*

*Proof.* Let $M \in \mathbb{R}^{n \times d}$ be any matrix whose $i$th row is comprised of i.i.d. random variables with distribution $\mathcal{N}(0, \sigma(i)^2)$. Define the row-wise (node-wise) quantization parameters $(s_i, z_i)$ for this row by

$$s_i = \frac{\max_{1 \le j \le d} M_{ij} - \min_{1 \le j \le d} M_{ij}}{q_{\max} - q_{\min}} \quad \text{and} \quad z_i = \min_{1 \le j \le d} M_{ij} - s_i\, q_{\min},$$

where $q_{\min}$ and $q_{\max}$ are fixed integers. The expectation of $\max_j M_{ij}$ and $\min_j M_{ij}$ can be expressed via the classical order-statistics integrals[2]. One has

$$\mathbb{E}\big[\max_{1 \le j \le d} M_{ij}\big] = \int_{-\infty}^{\infty} \Big[1 - F_X(x)\Big]^d dx,$$

where $F_X$ is the cumulative distribution function of $\mathcal{N}(0, \sigma(i)^2)$. $F_X$ is a continuous function of $\sigma(i)$. Also, this integral is absolutely convergent, implying that $\mathbb{E}[\max_j M_{ij}]$ is a continuous function of $\sigma(i)$. A similar argument shows that $\mathbb{E}[\min_j M_{ij}]$ is also continuous in $\sigma(i)$. Consequently,

$$\mathbb{E}[s_i] = g\big(\sigma(i)\big), \quad \mathbb{E}[z_i] = h\big(\sigma(i)\big),$$

for some continuous functions $g, h$. From Steps 1–3, we showed that the row variance $\sigma(i)^2$ in our GNN setting is proportional to a node-dependent scalar $\phi(v_i)$. Hence,

$$\mathbb{E}\big[s_i\big] = G\big(\phi(v_i)\big), \quad \mathbb{E}\big[z_i\big] = H\big(\phi(v_i)\big),$$

for some continuous functions $G, H$. Next, let

$$A = \max_{v \in V} \phi(v).$$

---

[2]As $d \to \infty$, it is known that

$$\frac{\mathbb{E}[\max_j M_{ij}]}{\sqrt{lnd}} \to \sqrt{2}\sigma$$

In other words, the integral is approximately $\sigma\sqrt{2\ln d}$ (Kamath, 2015).

Since $G, H$ are continuous on a compact set $[0, A]$, $G$ and $H$ are uniformly continuous on $[0, A]$. Therefore,

$$\left|\phi(u) - \phi(v)\right| < \delta \implies \left|G\big(\phi(u)\big) - G\big(\phi(v)\big)\right| < \epsilon \quad \text{and} \quad \left|H\big(\phi(u)\big) - H\big(\phi(v)\big)\right| < \epsilon,$$

for any desired $\epsilon > 0$. In particular, if $\phi(u) = \phi(v)$, then $\mathbb{E}[s_u] = \mathbb{E}[s_v]$ and $\mathbb{E}[z_u] = \mathbb{E}[z_v]$.

*Remark.* In the case that $M_{ij} = \mathrm{ReLU}\big(Y_{ij}\big)$ for i.i.d. zero mean Gaussian $Y_{ij}$, as $d \to \infty$, the probability of at least one entry being zero in each row goes to one, so $\mathbb{E}[\min_j \mathrm{ReLU}(Y_{ij})] \to 0$. Thus the above argument for scale parameters is valid for all matrices $X^{(l)}$ and $Z^{(l)} = X_c^{(l)}$.

This completes the proof. $\qquad\square$

## C DERIVATION OF TOPPIN FROM THEOREM 1.

In *Case 1* at Theorem 1, let $d(v)$ denote the indegree of node $v$, we can approximate all summations beyond the first term as a constant $C_1$, which yields:

$$\phi(v) = \sum_{v_{k_1} \in \mathcal{N}(v_v)} C_1 = d(v) \cdot C_1, \tag{12}$$

In *Case 2*, we approximate the summand of the second summation as a constant $C_2$, which yields:

$$\phi(v) = \frac{1}{d(v)} \sum_{v_{k_1} \in \mathcal{N}(v_v)} \left( \frac{1}{d(v_{k_1})^2} \sum_{v_{k_2} \in \mathcal{N}(v_{v_{k_1}})} C_2 \right) = \frac{1}{d(v)} \sum_{v_{k_1} \in \mathcal{N}(v_v)} \frac{C_2}{d(v_{k_1})} \tag{13}$$

This leads to the second element of TopPIN. Alternatively, approximating the entire summation as $C_3$ gives $\phi(v) \approx C_3/d(v)$, further reinforcing the choice of degree-based terms. This lightweight design effectively balances between accuracy and efficiency. Note that $\mathrm{TopPIN}(v)$ does not depend on the definition of $\tilde{A}$, and thus can be used for various GNNs. Empirically, we observed that the approximated first-order terms of TopPIN can capture most of the benefits with minimal overhead.

## D APPLICATION OF TOPGQ TO GAT-BASED ARCHITECTURES

We provide a theoretical justification for applying TopPIN in GAT-based architectures, which operates on edge weights obtained via the softmax function. We analyze the bound of *expected variance of node activations*, and show that the *expected variance of GAT node activation is bounded by the terms of TopPIN*. We demonstrate that these bounds align with the structure of TopPIN, validating its use as a lightweight proxy for per-node quantization.

### D.1 SETUP AND ASSUMPTIONS

We consider a GAT layer as defined in Section A:

$$X^{(l+1)} = ReLU\big( \tilde{A} X^{(l)} W^{(l)} \big), \quad l = 0, \dots, L-1,$$
$$Z^{(l)} = X^{(l)} W^{(l)}, \quad Y^{(l)} = \tilde{A} Z^{(l)}, \quad X^{(l+1)} = ReLU\big(Y^{(l)}\big).$$

with the following assumptions, in line with Section A:

- $Z_{ij}^{(0)} \sim \mathcal{N}(0, \ d_0 \sigma_x^2 \sigma_0^2)$ (i.i.d. per feature dimension),
- Assume edge weights for target node $i$ are drawn from a normal distribution, specifically $\alpha_i \sim \mathcal{N}(1/d(i), \sigma_{\alpha_i}^2)$, independently across $j$ and independently across layers, where

$$0 < \sigma_{\alpha_i}^2 \le \frac{1}{d(i)} - \frac{1}{d(i)^2}.$$

We derive the upper bound of $\sigma_\alpha^2$ as follows:

Since $\sum_j \alpha_{ij} = 1$, and $\alpha_{ij} \ge 0$, $\mathbb{E}[\alpha_i] = \frac{1}{d(i)}$. Using the Cauchy-Schwarz inequality, $\mathbb{E}[\alpha_i^2] \ge (\mathbb{E}[\alpha_i])^2$ and the bound $0 \le \alpha_{ij} < 1$, this leads to the bound $\frac{1}{(d(i))^2} \le \mathbb{E}[\alpha_i^2] \le \mathbb{E}[\alpha_i] = \frac{1}{d(i)}$.

$$\text{therefore, } 0 < \sigma_{\alpha_i}^2 = \mathrm{Var}[\alpha_{ij}] = \mathbb{E}[\alpha_{ij}^2] - (\mathbb{E}[\alpha_{ij}])^2 \le \frac{1}{d(i)} - \frac{1}{(d(i))^2}$$

## D.2 DERIVING THE ACTIVATION VARIANCE BOUND

We aim to quantify the variance of the output activation $Y_i^{(l)}$ at node $i$. Consider a coordinate $j$:

$$Y_{ij}^{(0)} = \sum_{k \in \mathcal{N}(i)} \alpha_{ik} Z_{kj}^{(0)}.$$

Since $Z_{kj}^{(0)} \sim \mathcal{N}(0,\ d_0 \sigma_x^2 \sigma_0^2)$ are independent, The mean and variance of $Y_{ij}^{(0)}$ is:

$$\mathbb{E}[Y_{ij}^{(0)}] = \sum_{k=1}^{d(i)} \mathbb{E}[\alpha_{ik} Z_{kj}^{(0)}] = 0, \tag{14}$$

$$\mathrm{Var}(Y_{ij}^{(0)}) = \sum_{k=1}^{d(i)} \mathbb{E}[\alpha_{ik}^2] \cdot \mathbb{E}[(Z_{kj}^{(0)})^2] \tag{15}$$

$$= (d(i))\left(\mathrm{Var}(\alpha_i) + (\mathbb{E}[\alpha_i])^2\right) \cdot d_0 \sigma_x^2 \sigma_0^2 = (d(i))\left(\sigma_{\alpha_i}^2 + \frac{1}{(d(i))^2}\right) \cdot d_0 \sigma_x^2 \sigma_0^2 \tag{16}$$

Denote $\psi(i)$ as $\left(\sigma_{\alpha_i}^2 + \frac{1}{(d(i))^2}\right)$. We can bound $\psi(i)$ with $\sigma_{\alpha_i}^2$ as below.

$$\frac{1}{(d(i))^2} < \psi(i) = \left(\sigma_{\alpha_i}^2 + \frac{1}{(d(i))^2}\right) \leq \frac{1}{(d(i))}, \quad \frac{1}{(d(i))} < \mathrm{Var}(Y_{ij}^{(0)}) \leq 1$$

By the Central Limit Theorem (CLT), $Y_{ij}^{(0)}$ is approximately $\mathcal{N}(0,\ d_0 \sigma_x^2 \sigma_0^2 \cdot d(i)\psi(i))$. Since quantization scale (e.g., in min–max quantization) is influenced by the activation variance, bounding $\psi(i)$ leads directly to bounding the expected quantization parameters. We show that the lower bound of $\psi(i)$ is equivalent to the first term of TopPIN.

With the same assumption and operation for the next layer, and $Y^{(0)}$ as input, the approximated distribution of $Y_{ij}^{(1)}$ is $\mathcal{N}(0,\ c \cdot \psi_2(i))$, with c as a constant, and $\psi_2(i)$ bounded in the range of $(\frac{1}{(d(i))^2}(\sum_{k \in \mathcal{N}(i)} \frac{1}{d(k)}), 1]$. We demonstrate that the lower bound of $\psi_2(v)$ is equivalent to the product of each term in TopPIN, bringing a strong correlation to the formulation of TopPIN.

## D.3 TOPPIN AS A PROXY FOR FEATURE VARIANCE

$$\mathrm{TopPIN}(v) = \left(d(v),\ \frac{1}{d(v)}\sum_{u \in \mathcal{N}(v)} \frac{1}{d(u)}\right).$$

Since $\psi(v)$ is a degree-dependent property, *TopPIN provides a topology-aware approximated bound* of the per-node activation statistics, without computing the attention scores explicitly.

GAT activation variance is governed by $\psi(v)$. Modeling $\alpha_{ij} \sim \mathcal{N}(1/d(i), \sigma_{\alpha_i}^2)$ gives a tight, degree-bounded expectation for $\psi(v)$. Quantization parameters are thus bounded in expectation by $d(v)$. TopPIN aligns with these bounds and provides a practical, theoretically grounded proxy for quantization in GNNs with softmax-generated edge weights.

# E ADDITIONAL EXPERIMENTAL RESULTS

## E.1 EXPERIMENTAL RESULTS OF GRAPHSAGE ARCHITECTURE ON INDUCTIVE SETTING

We evaluate the quantization accuracy and quantization speed of our method (TopGQ) on the Graph-SAGE architecture (Table 8), covering node classification (Cora, Citeseer, PubMed) and graph classification tasks (IMDB-BINARY, COLLAB). The experimental results show that TopGQ consistently achieves competitive or superior accuracy compared to baseline methods, while significantly reducing quantization time. These results further support that TopGQ generalizes effectively across architectures, providing acceleration for GraphSAGE models while achieving comparable performance to quantization-aware training methods that require substantially longer quantization times.

Table 8: Comparison of quantization accuracy with GraphSAGE architecture

| Method | Node Classification | | | | | | Graph Classification | | | |
| | Cora | | Citeseer | | PubMed | | IMDB-B | | COLLAB | |
| | Acc. | Q.Time | Acc. | Q.Time | Acc. | Q.Time | Acc. | Q.Time | Acc. | Q.Time |
|---|---|---|---|---|---|---|---|---|---|---|
| FP32 | 77.02 | - | 76.34 | - | 89.18 | - | 77.46 | - | 80.36 | - |
| **INT8** | | | | | | | | | | |
| SGQ | 76.28 | (8.24s) | 76.06 | (14.84s) | 88.87 | (23.23s) | 66.36 | (6.15m) | 80.29 | (38.21m) |
| DQ | 75.50 | (23.87s) | 74.77 | (69.95s) | 88.62 | (3.07m) | 73.04 | (9.09m) | 79.32 | (2.28h) |
| EPQ | 72.84 | (21.16s) | 74.44 | (73.24s) | 84.70 | (1.80m) | 71.91 | (14.02m) | 79.78 | (2.51h) |
| $A^2Q$ | 76.94 | (4.56s) | 75.08 | (4.96s) | 88.76 | (18.21s) | 74.49 | (4.97m) | 79.64 | (14.59m) |
| QLR | **77.96** | (4.12s) | 31.78 | (4.62s) | 88.08 | (7.28s) | 63.42 | (3.67s) | 70.16 | (13.71m) |
| DRA | 76.46 | (3.11s) | 75.74 | (3.00s) | 88.98 | (3.15s) | 76.51 | (2.67m) | 80.27 | (12.12m) |
| TopGQ | 76.86 | (0.54s) | **76.32** | (0.56s) | **89.00** | (0.62s) | **77.23** | (2.93s) | **80.53** | (15.89s) |
| **INT4** | | | | | | | | | | |
| SGQ | 75.52 | (8.41s) | **75.94** | (14.65s) | 86.62 | (23.24s) | 65.56 | (6.11m) | 78.30 | (39.20m) |
| DQ | 74.36 | (23.49s) | 74.99 | (69.91s) | 88.58 | (3.07m) | 73.52 | (9.10m) | 79.02 | (2.26h) |
| EPQ | 73.00 | (21.10s) | 74.58 | (73.47s) | 84.44 | (1.80m) | 61.00 | (14.06m) | 58.92 | (2.58h) |
| $A^2Q$ | 74.66 | (4.65s) | 73.00 | (5.01s) | 85.32 | (18.17s) | 73.92 | (4.93m) | 66.12 | (14.64m) |
| QLR | 74.52 | (4.05s) | 30.68 | (4.43s) | **87.42** | (7.29s) | 63.30 | (4.69s) | 63.30 | (13.62m) |
| DRA | 76.18 | (3.24s) | 74.60 | (2.93s) | 78.84 | (3.42s) | 75.04 | (2.56m) | 78.18 | (12.12m) |
| TopGQ | **76.30** | (0.53s) | 75.76 | (0.57s) | 87.26 | (0.62s) | **75.44** | (2.91s) | **79.38** | (15.92s) |

∗SGQ: SGQuant, DQ: Degree-Quant, EPQ: EPQquant

Table 9: Comparison of quantization accuracy on transductive setting

| Dataset | Method | INT4 | | | | | | | | INT8 | | | | | | | |
| | | GCN | | GAT | | GIN | | GraphSAGE | | GCN | | GAT | | GIN | | GraphSAGE | |
| | | Acc. | Q.Time | Acc. | Q.Time | Acc. | Q.Time | Acc. | Q.Time | Acc. | Q.Time | Acc. | Q.Time | Acc. | Q.Time | Acc. | Q.Time |
|---|---|---|---|---|---|---|---|---|---|---|---|---|---|---|---|---|---|
| Cora | FP32 | 87.72 | - | 88.08 | - | 86.14 | - | 85.70 | - | 87.72 | - | 88.08 | - | 86.14 | - | 85.70 | - |
| | SGQ | **87.46** | (4.09s) | 82.32 | (7.09s) | 78.92 | (4.46s) | 85.82 | (6.33s) | **87.88** | (4.18s) | **88.14** | (7.05s) | 86.02 | (4.46s) | 85.94 | (6.37s) |
| | DQ | 86.40 | (8.96s) | 87.10 | (11.72s) | 83.10 | (29.46s) | 86.04 | (33.37s) | 87.12 | (9.77s) | 87.54 | (11.85s) | 80.34 | (30.50s) | 86.60 | (33.50s) |
| | EPQ | 83.44 | (40.74s) | 86.50 | (42.17s) | 41.96 | (40.89s) | 84.96 | (40.66s) | 86.50 | (40.71s) | 86.98 | (42.09s) | 81.20 | (40.77s) | 84.80 | (40.60s) |
| | $A^2Q$ | 55.70 | (1.99s) | 75.80 | (3.65s) | 85.30 | (2.45s) | **86.20** | (2.48s) | 87.40 | (2.06s) | 87.60 | (3.72s) | 86.10 | (2.47s) | **87.20** | (2.75s) |
| | QLR | 85.20 | (2.47s) | 87.04 | (3.88s) | 85.30 | (2.44s) | 85.78 | (2.16s) | 86.54 | (2.41s) | 87.02 | (3.73s) | 84.88 | (2.30s) | 86.74 | (2.10s) |
| | DRA | 82.60 | (1.02s) | 77.58 | (2.51s) | 27.52 | (0.98s) | 83.80 | (1.10s) | 87.56 | (1.04s) | 87.84 | (2.48s) | 85.40 | (1.06s) | 85.58 | (1.09s) |
| | TopGQ | 87.38 | (0.66s) | 87.64 | (0.46s) | 85.86 | (0.51s) | 85.90 | (0.46s) | 87.78 | (0.58s) | 88.10 | (0.65s) | 86.18 | (0.55s) | 86.10 | (0.75s) |
| Citeseer | FP32 | 79.84 | - | 79.78 | - | 79.36 | - | 79.56 | - | 79.84 | - | 79.78 | - | 79.36 | - | 79.56 | - |
| | SGQ | 79.10 | (6.50s) | 79.22 | (9.36s) | 77.28 | (8.56s) | 79.36 | (12.64s) | **79.88** | (6.40s) | 80.02 | (9.46s) | **79.54** | (8.60s) | 79.52 | (12.55s) |
| | DQ | 24.32 | (21.36s) | 23.10 | (23.67s) | 70.58 | (85.98s) | 23.10 | (99.04s) | 79.56 | (21.14s) | 79.72 | (23.87s) | 72.24 | (86.44s) | 78.98 | (100.53s) |
| | EPQ | 78.78 | (137.88s) | 79.34 | (139.37s) | 47.22 | (138.38s) | 77.26 | (138.65s) | 79.14 | (138.02s) | 79.36 | (139.43s) | 70.60 | (138.41s) | 78.24 | (138.64s) |
| | $A^2Q$ | 53.90 | (2.20s) | 64.00 | (3.92s) | 78.30 | (4.28s) | 78.60 | (5.47s) | 76.50 | (2.21s) | 79.80 | (3.90s) | 79.50 | (4.27s) | 79.20 | (5.53s) |
| | QLR | 67.78 | (2.54s) | **79.72** | (3.92s) | 75.74 | (3.72s) | 76.16 | (4.60s) | 77.82 | (2.62s) | 79.24 | (3.89s) | 75.00 | (3.74s) | 79.20 | (4.39s) |
| | DRA | 77.56 | (1.63s) | 78.88 | (2.46s) | 42.34 | (1.63s) | 78.90 | (1.53s) | 79.70 | (1.63s) | 79.78 | (1.63s) | 79.38 | (1.61s) | **79.70** | (1.50s) |
| | TopGQ | 79.56 | (0.46s) | 79.48 | (0.51s) | 79.26 | (0.61s) | 79.98 | (0.66s) | 79.86 | (0.46s) | 79.82 | (0.62s) | 79.44 | (0.55s) | 79.68 | (0.63s) |
| PubMed | FP32 | 88.36 | – | 87.76 | – | 89.42 | – | 89.38 | – | 88.36 | – | 87.76 | – | 89.42 | – | 89.38 | – |
| | SGQ | 86.52 | (12.64s) | 82.86 | (6.34s) | 86.02 | (10.18s) | 88.84 | (8.70s) | **88.64** | (6.30s) | 87.50 | (10.11s) | 89.72 | (8.60s) | **89.72** | (11.66s) |
| | DQ | 87.26 | (99.04s) | 87.50 | (20.27s) | 88.60 | (33.37s) | 88.84 | (105.63s) | 88.02 | (19.28s) | 87.04 | (31.99s) | 89.54 | (103.28s) | 88.84 | (94.59s) |
| | EPQ | 84.34 | (138.65s) | 86.08 | (101.95s) | 52.32 | (2.90s) | 87.06 | (2.07s) | 85.40 | (102.00s) | 86.54 | (103.37s) | 86.90 | (102.50s) | 87.16 | (102.61s) |
| | $A^2Q$ | 79.70 | (5.47s) | 82.40 | (102.52s) | 89.10 | (107.55s) | 87.20 | (105.27s) | 87.20 | (2.16s) | 87.10 | (6.95s) | **90.30** | (4.94s) | 88.70 | (5.42s) |
| | QLR | 79.48 | (4.60s) | 87.42 | (2.59s) | 85.40 | (3.73s) | 88.86 | (4.23s) | 87.06 | (2.50s) | 87.40 | (3.90s) | 88.46 | (4.21s) | 88.94 | (4.49s) |
| | DRA | 85.10 | (1.47s) | 83.26 | (2.82s) | 47.30 | (1.42s) | 85.96 | (2.42s) | 88.28 | (1.43s) | 87.70 | (2.81s) | 88.76 | (1.45s) | 89.20 | (2.43s) |
| | TopGQ | 87.72 | (0.66s) | 87.52 | (0.50s) | 89.20 | (0.49s) | 88.92 | (0.86s) | 88.42 | (0.52s) | 87.86 | (0.53s) | 89.40 | (0.62s) | 89.14 | (0.56s) |

∗SGQ: SGQuant, DQ: Degree-Quant, EPQ: EPQuant

## E.2 NODE CLASSIFICATION ON TRANSDUCTIVE SETTING

We report additional experimental results on transductive node classification tasks in Table 9, in addition to the inductive results presented in Table 1. The experimental results in Table 9 consistently align with the trends that TopGQ persistently achieves the lowest quantization times compared to

Table 10: Comparison of quantization accuracy on large scale inductive datasets

| Method | INT4 | | | | | | | | INT8 | | | | | | | |
| --- | --- | --- | --- | --- | --- | --- | --- | --- | --- | --- | --- | --- | --- | --- | --- | --- |
| | ogbn-arxiv | | | | ogbn-proteins | | | | ogbn-arxiv | | | | ogbn-proteins | | | |
| | GCN | | GraphSAGE | | GCN | | GraphSAGE | | GCN | | GraphSAGE | | GCN | | GraphSAGE | |
| | Acc. | Q.Time | Acc. | Q.Time | R-A | Q.Time | R-A | Q.Time | Acc. | Q.Time | Acc. | Q.Time | R-A | Q.Time | R-A | Q.Time |
| FP32 | 55.57 | - | 57.54 | - | 68.96 | - | 72.00 | - | 55.57 | - | 57.54 | - | 68.96 | - | 72.00 | - |
| SGQ | 55.25 | (1.24**m**) | 57.65 | (1.87**m**) | 70.75 | (8.93**m**) | 73.18 | (4.44**m**) | 24.03 | (1.22**m**) | 52.90 | (1.88**m**) | 54.50 | (8.90**m**) | 65.69 | (4.44**m**) |
| DQ | 55.67 | (9.30**m**) | 57.46 | (13.70**m**) | 60.08 | (37.59**m**) | 73.41 | (24.54**m**) | 54.02 | (8.88**m**) | 56.83 | (13.86**m**) | 52.89 | (36.90**m**) | 71.84 | (23.82**m**) |
| EPQ | 43.71 | (5.61**m**) | 46.72 | (5.65**m**) | 63.26 | (4.96**m**) | 58.65 | (2.85**m**) | 27.11 | (5.60**m**) | 44.82 | (5.65**m**) | 52.87 | (4.96**m**) | 57.71 | (2.84**m**) |
| $A^2$Q | 47.06 | (37.39**s**) | 57.51 | (48.33**s**) | 49.83 | (4.42**m**) | 72.18 | (2.80**m**) | 24.00 | (36.55**s**) | 55.24 | (46.69**s**) | 47.99 | (4.41**m**) | 70.06 | (2.79**m**) |
| QLR | 55.54 | (46.06**s**) | 57.46 | (57.17**s**) | 65.76 | (7.11**m**) | 73.65 | (3.07**m**) | 52.96 | (45.19**s**) | 55.98 | (57.60**s**) | 56.21 | (7.15**m**) | 50.26 | (3.07**m**) |
| DRA | 54.46 | (25.55**s**) | 57.55 | (31.99**s**) | 56.91 | (3.47**m**) | 72.15 | (1.68**m**) | 22.76 | (25.47**s**) | 53.56 | (31.81**s**) | 51.73 | (3.43**m**) | 63.89 | (1.67**m**) |
| TOPGQ | 55.86 | (0.52**s**) | 57.55 | (0.61**s**) | 68.50 | (1.28**s**) | 73.07 | (1.27**s**) | 47.97 | (0.51**s**) | 54.48 | (0.59**s**) | 60.39 | (1.28**s**) | 70.72 | (1.26**s**) |

∗R-A: ROC-AUC, SGQ: SGQuant, DQ: Degree-Quant, EPQ: EPQuant

Table 11: Comparison of quantization accuracy on molecular-domain datasets

| Method | INT4 | | | | | | | | INT8 | | | | | | | |
| --- | --- | --- | --- | --- | --- | --- | --- | --- | --- | --- | --- | --- | --- | --- | --- | --- |
| | MUTAG | | | | PPI | | | | MUTAG | | | | PPI | | | |
| | GCN | | GraphSAGE | | GCN | | GraphSAGE | | GCN | | GraphSAGE | | GCN | | GraphSAGE | |
| | Acc. | Q.Time | Acc. | Q.Time | Acc. | Q.Time | Acc. | Q.Time | Acc. | Q.Time | Acc. | Q.Time | Acc. | Q.Time | Acc. | Q.Time |
| FP32 | 87.44 | – | 86.74 | – | 71.10 | – | 91.37 | – | 87.44 | – | 86.74 | – | 71.10 | – | 91.37 | – |
| SGQ | 81.57 | (86.27**s**) | **86.02** | (100.65**s**) | 55.71 | (86.27**m**) | 59.46 | (100.65**m**) | 81.87 | (84.92**s**) | 83.53 | (102.82**s**) | **73.26** | (6.74**m**) | **91.33** | (39.35**m**) |
| DQ | 85.27 | (109.07**s**) | 80.66 | (104.39**s**) | 48.32 | (9.08**m**) | 64.24 | (2.39**m**) | 86.00 | (105.04**s**) | 80.24 | (102.78**s**) | 64.55 | (9.08**m**) | 67.17 | (2.39**h**) |
| EPQ | 76.11 | (52.22**s**) | 76.58 | (55.83**s**) | 51.18 | (14.17**m**) | **74.71** | (2.99**m**) | 78.27 | (51.20**s**) | 77.11 | (55.13**s**) | 63.87 | (14.17**m**) | 81.59 | (2.99**h**) |
| $A^2$Q | 80.99 | (47.06**s**) | 77.99 | (55.59**s**) | 40.83 | (3.78**m**) | 43.12 | (14.75**m**) | 79.30 | (45.56**s**) | 83.12 | (50.89**s**) | 40.46 | (3.78**m**) | 45.78 | (14.75**m**) |
| QLR | **89.88** | (51.85**s**) | 79.27 | (51.90**s**) | **61.43** | (3.70**m**) | 58.60 | (13.04**m**) | **86.64** | (50.25**s**) | 76.08 | (54.26**s**) | 62.49 | (3.70**m**) | 71.89 | (13.04**m**) |
| DRA | 85.78 | (33.14**s**) | 85.22 | (37.07**s**) | 33.29 | (2.29**m**) | 49.74 | (11.49**m**) | 85.86 | (32.30**s**) | 83.60 | (34.89**s**) | **73.26** | (2.29**m**) | 88.00 | (11.49**m**) |
| TOPGQ | 78.30 | (**1.29s**) | 82.68 | (**1.22s**) | 61.10 | (**1.87s**) | 71.53 | (**13.86s**) | 86.61 | (**1.29s**) | **86.74** | (**5.58s**) | 73.48 | (**1.23s**) | 92.54 | (**13.86s**) |

∗SGQ: SGQuant, DQ: Degree-Quant, EPQ: EPQuant

baselines, while maintaining comparable or superior accuracy. This advantage in quantization speed demonstrates the practical value and effectiveness of TopGQ, particularly in resource-constrained environments. We show that this advantage holds in both transductive and inductive settings.

### E.3 EXPERIMENTAL RESULTS ON LARGE SCALE INDUCTIVE DATASETS.

To further evaluate the generalizability of TopGQ, we provide additional quantization results on ogbn-arxiv and ogbn-proteins in Table 10, both inductive node classification tasks. For these experiments, we use GCN and GraphSAGE as baseline models, the same GNN baseline architectures used for the original paper for the dataset (Hu et al., 2020). Compared to existing methods, TopGQ achieves comparable or superior performance without requiring retraining or gradient-based updates. Also, TopGQ acceleration gains in the quantization time is up to $92 \times -1,076\times$ compared to baseline methods. These results further demonstrate the effectiveness of TopGQ and its ability to generalize across diverse datasets.

## F LIMITATIONS OF TOPGQ

In this section, we analyze the limitations of TopGQ. First, we find that TopGQ shows limited performance when applied to molecule-domain datasets. We report the results on molecule-domain datasets in Table 11, using MUTAG and PPI (Zitnik & Leskovec, 2017). In the table, we can observe a more noticeable gap between other baselines and TopGQ. We believe this partially comes from the two reasons: the lack of topological diversity, and the heterophily between connected nodes. Since each graph is a molecular chain, the nodes exhibit a short range of degree diversity and have weak distinguishability in topology. As our work builds on the distinct topological characteristics of the graph, it has a limited advantage in such weak-topology graphs. Also, the graphs in the molecular domain tend to have a heterophilic connection, as a vast amount of edges connect to different molecules. To overcome such limitations, we restrict parameter sharing to nodes with matching input features and similar TopPIN values, thereby encoding heterophily. On the PPI dataset, this strategy proved effective in preserving accuracy, with additional gains up to $2.86\% - 9.47\%$.

Table 12: Comparison of theoretical costs and storage for different methods.

| Metrics | Theoretical Cost | Theoretical Storage |
|---|---|---|
| FP32 | $O_{FP}(N^2 F_1 + N F_1 F_2)$ | $O_{FP}(E + F_1 F_2 + N F_0)$ |
| Degree-Quant | $O_{INT}(N^2 F_1 + N F_1 F_2) + O_{FP_{elem}}(N F_2)$ | $O_{INT}(E + F_1 F_2 + N F_0) + O_{FP}(1)$ |
| Degree-Quant-PTQ | $O_{INT}(N^2 F_1 + N F_1 F_2) + O_{FP_{elem}}(N F_2)$ | $O_{INT}(E + F_1 F_2 + N F_0) + O_{FP}(1)$ |
| TopGQ | $O_{INT}(N^2 F_1 + N F_1 F_2) + O_{FP_{elem}}(N F_2)$ | $O_{INT}(E + F_1 F_2 + N F_0) + O_{FP}(N_T + F_2)$ |

Secondly, we find that dual-axis scale absorption introduces an additional runtime operation when quantizing GNN architectures with dynamic edge weights. In such cases, we fuse precalculated scaling vectors to the quantization scale matrix for the adjacency matrix, so that dual-axis scale absorption operation itself can be fused with the runtime quantization of runtime-calculated edge weights. The absorption will alter the SPMV operation between scale vectors and edge weights to SPMM operation between scale matrices and edge weights when the absorption is fused with the online computation. Although this may slightly increase the floating-point operations of runtime-computed edge weights, the online quantization of arbitrary edge weights is a global overhead across all GNN quantization methods (Feng et al., 2020; Tailor et al., 2020; Huang et al., 2022; Zhu et al., 2022; Wang et al., 2023; Jeddi et al., 2024), which also target to quantize GNNs with dynamic edge weights. On top of that, we believe the difference of the inference time can be fairly negligible by parallelism within GPU operations.

## G  QUANTIZATION TRADE-OFF AND COMPRESSION ANALYSIS OF TOPGQ

Here, we present a comprehensive analysis regarding the trade-offs and compression advantages of TopGQ. We provide analysis of computational cost and storage consumption. The theoretical analysis is shown in Table 12.

TopGQ finds a good balance between reducing quantization time and preserving accuracy, while other choices in FP32, Degree-Quant, TopGQ demonstrate disadvantages in either accuracy, time, or memory. FP32 suffers from the expensive costs of computation and storage. While Degree-Quant alleviates this cost via quantization, the long quantization time is required to obtain the benefits. TopGQ is free from the quantization time problem but at the cost of considerable performance degradation. TopGQ aims to find the best way of addressing each issue by leveraging topological node similarities with an additional amount of storage cost.

As for the theoretical costs (Table 12), we assume GNN layer propagation as AXW operation, with $A \in \mathbb{R}^{N \times N}, X \in \mathbb{R}^{N \times F1}, W \in \mathbb{R}^{F1 \times F2}$ with initial dataset size of $N \times F_0$. We note the computation and storage costs of floating-point (FP) and integer (INT) operations as follows:

- $O_{FP}()$: Complexity for FP operations / Storage complexity for FP values.
- $O_{FP_{elem}}()$: Complexity for element-wise FP operations.
- $O_{INT}()$: Complexity for INT operations / Storage complexity for INT values.

The computational cost shows that quantization converts the expensive floating-point matrix multiplication into integer operations. The additional floating-point cost comes from converting integer outputs back to floating-point values. The theoretical analysis is based on (Zhu et al., 2022).

To further validate the actual compression advantage, we provide the results of memory usage reduction ratios for inference components of a GCN model on Reddit dataset at Table 13. This confirms that TopGQ can effectively benefit from model/data memory reduction and faster inference.

## H  DATASET STATISTICS

We report the dataset statistics used for the evaluation of our method, TopGQ at Table 14. To assess generalizability, we selected datasets spanning a range of scales. Note that we evaluate graph-level datasets with 10-fold cross-validation, with a fixed validation/test set size per fold.

Table 13: Actual quantization reduction ratios for INT8 and INT4

| Inference Component | INT8 | INT4 |
|---|---|---|
| Graph Input (Node Features) | 3.995× | 7.982× |
| Model Intermediate Activation | 3.987× | 7.951× |
| Model Weights | 3.922× | 7.681× |
| Total Reduction Ratio | 3.992× | 7.971× |

Table 14: Statistics of node-level and graph-level datasets for evaluation.

| Node-Level Datasets | Graph # | Node # | Edge # | Train Node # | Val Node # | Test Node # | Class # |
|---|---|---|---|---|---|---|---|
| Cora | 1 | 2,708 | 10,556 | 140 | 500 | 1,000 | 7 |
| Citeseer | 1 | 3,327 | 9,104 | 120 | 500 | 1,000 | 6 |
| Pubmed | 1 | 19,717 | 88,648 | 60 | 500 | 1,000 | 3 |
| PPI | 24 | 56,944 | 1,587,264 | 44,906 | 6,514 | 5,524 | 121 |
| Reddit | 1 | 232,965 | 114,615,892 | 153,431 | 23,831 | 55,703 | 41 |
| ogbn-proteins | 1 | 132,534 | 79,122,504 | 86,619 | 21,236 | 24,679 | 112 |
| ogbn-arxiv | 1 | 169,343 | 1,166,243 | 90,941 | 29,799 | 48,603 | 40 |
| ogbn-products | 1 | 2,449,029 | 123,718,280 | 196,615 | 39,323 | 2,213,091 | 47 |
| MAG240M | 1 | 244,160,499 | 1,728,364,232 | 1,112,392 | 138,949 | 88,092 | 153 |

| Graph-Level Datasets | Graph # | Avg. Node # | Avg. Edge # | Train Graph # | Val Graph # | Test Graph # | Class # |
|---|---|---|---|---|---|---|---|
| MUTAG | 188 | 17.9 | 39.6 | 150 | 19 | 19 | 2 |
| IMDB-BINARY | 1,000 | 19.8 | 193.1 | 800 | 100 | 100 | 2 |
| COLLAB | 5,000 | 74.5 | 4914.4 | 4,000 | 500 | 500 | 3 |

# I  ADDITIONAL EXPERIMENTAL SETTINGS

We report evaluation results on two representative graph processing tasks: Node-level classification, graph-level classification. For node-level classification, we compare the accuracy of Cora, Citeseer, PubMed, Reddit, ogbn-products, and MAG240M in inductive setting. For the inductive setting, we construct a training graph containing only train nodes and separate validation/test graphs containing only validation or test nodes For graph-level classification, we choose IMDB-BINARY and COL-LAB datasets to evaluate the inductive inference performance of quantized GNNs. We report the accuracy by 10-fold cross-validation, with a fixed random seed.

All experiments are conducted and measured on a server with a single A6000 GPU, RTX 4090 GPU, and Intel(R) Xeon(R) Gold 6442Y CPU. We implement our algorithm on PyG library v2.5.2 with PyTorch v2.4.0. In the index computation, we use the SciPy library and Pytorch implementations.

# J  CODE

The code, which includes our implementation of this work, is included in a zip archive of the supplementary material. The code is under GNU General Public License v3.0. The guideline to run the code and reproduce the results from TopGQ is provided in the README file.

