# OpenReview forum: "TopGQ: Fast GNN Post-Training Quantization Leveraging Topology Information"
_ICLR.cc/2026/Conference — ICLR 2026 Conference Withdrawn Submission_

### Official Review · Reviewer_gi3j · 2025-10-29

**Soundness:** 2
**Presentation:** 3
**Contribution:** 2
**Rating:** 4
**Confidence:** 3

**Summary:**

This paper presents TopGQ, a post-training quantization framework for GNNs that exploits graph topology to achieve fast quantization. The method introduces two main techniques: dual-axis scale absorption and TopPIN. The authors claim the speedup in quantization time compared to existing methods while maintaining comparable accuracy across various node and graph classification tasks.

**Strengths:**

1. The paper tackles a genuine bottleneck in GNN deployment - the excessive quantization time of existing methods (hours to days for large graphs) that limits practical applicability.

2. The authors provide theoretical justification (Theorems 1-2) linking local graph topology to activation variance, which motivates the use of topology-aware quantization.

3. Comprehensive experiments: Extensive evaluation across multiple architectures (GCN, GAT, GIN, GraphSAGE), datasets of varying scales (from Cora to MAG240M with 240M nodes), and both node/graph-level tasks.

**Weaknesses:**

1.  The claim “quantizing GNNs is known to be difficult due to the varying magnitudes of node activations” lacks empirical support. Is this issue universally observed across all GNN architectures? If the variation arises from graph structure, the use of normalized adjacency matrices could alleviate it. If it instead results from activation outliers, quantitative evidence should be provided—e.g., what proportion of nodes are considered outliers?

2. A major concern lies in the applicability of the proposed techniques. Dual-axis scale absorption assumes a static adjacency matrix, whereas many recent GNNs employ dynamic or learnable edge weights. Although the authors briefly acknowledge this limitation in the appendix, the quantitative impact of the additional cost remains unclear. For example, in classical graph transformers—where edge weights are computed for all node pairs—what is the precise computational overhead introduced by the proposed absorption?

3. The motivation behind several design choices is insufficiently explained. For instance, the reasoning behind Formula (10) is unclear and should be elaborated to justify its formulation.

4. Theoretical results rely on assumptions that are unrealistic in practice—for example, Theorem 1 assumes infinitely large hidden dimensions and zero-mean inputs and weights. These assumptions weaken the practical validity of the theoretical claims.

5. Since the proposed method is a PTQ approach, comparing it primarily against QAT baselines is not entirely fair or informative. The inclusion of more recent PTQ baselines would strengthen the experimental section.

6. What are the numbers in Table 7? Besides, the numerical results in Table 7 show substantial improvement, but the underlying reasons for such gains are not explained. A discussion on why these improvements occur is needed.

7. What does “GS” denote—GraphSAGE? If so, do the authors have results on more recent GNNs or graph transformer architectures to demonstrate broader applicability?

8. DRA achieves only 1.75 % accuracy on Reddit GCN INT4 and 3.12 % on ogbn-products GCN INT4. Were these results reproduced by the authors or directly taken from the original publication?

9. The selection of k from the k nearest neighbors (mentioned in lines 232–237 and Figure 3c) are not specified. These details are crucial for reproducibility.

10. The reported speed-up in Table 5 appears modest compared to the claimed efficiency gains. Additional clarification or discussion is required to reconcile this discrepancy.

11. The paper focuses on the computation time speed up. However, no empirical results of memory cost are provided.  Will the proposed method introduce additional memory cost?

**Questions:**

Please refer to weakness.

---

### Official Review · Reviewer_DDaU · 2025-10-31

**Soundness:** 2
**Presentation:** 3
**Contribution:** 2
**Rating:** 2
**Confidence:** 4

**Summary:**

This paper proposes TopGQ, a topology-aware post-training quantization (PTQ) framework for GNNs. The motivation is that existing GNN quantization methods are time consuming.

TopGQ introduces two main components:
	1.	Dual-Axis Scale Absorption, which merges node-wise scaling factors into the adjacency matrix, enabling integer arithmetic compatibility while maintaining per-node quantization precision.
	2.	TopPIN (Topology-Aware Pairwise Index), a lightweight node index derived from local graph structure (node degree and neighbor degree) that predicts activation variance and allows quantization parameter reuse for unseen nodes in inductive settings.

Experiments across diverse GNNs and datasets show that TopGQ does reduce quantization time  while matching or exceeding sota accuracy.
Theoretical sections attempt to connect local topology to node activation variance and provide continuity arguments for quantization parameter sharing.

**Strengths:**

The experiments cover enough datasets and hardware setups( desktop GPUs and edge-devices)

Even if the industrial motivation is overstated, the efficiency gains would still be useful for frequent retraining or memory-constrained deployment.

**Weaknesses:**

The cited works discuss dynamic GNN training/inference, not quantization time bottlenecks. Thus, the paper’s claim that “quantization time is a deployment barrier” lacks direct evidence.

Dual-axis absorption is somewhat a reparameterization trick (scaling merge and Hadamard product rewrite).

Theoretical analysis of TopPIN relies on strong assumptions without providing error bounds or practical ablations verifying these assumptions.

**Questions:**

Find a stronger motivation for effient Post-Training GNN Quantization.

---

### Official Review · Reviewer_2HBD · 2025-11-02

**Soundness:** 2
**Presentation:** 3
**Contribution:** 2
**Rating:** 4
**Confidence:** 5

**Summary:**

This paper introduces TopGQ, a topology-aware post-training quantization framework for graph neural networks. Its core contributions include: a dual-axis scale absorption technique that maintains the efficiency of integer arithmetic while achieving node-wise quantization accuracy, and a lightweight topological index (TopPIN) designed to rapidly predict quantization parameters for unseen nodes. TopGQ aims to provide a quantization solution for large-scale GNN deployment; however, several concerns regarding its novelty, methodology, and experimental evaluation require further analysis.

**Strengths:**

1. This paper proposes an interesting idea, supported by a discussion.

2. This paper exhibits a clear structure and is well-written.

**Weaknesses:**

This paper lacks enough innovation.

1. The core of the proposed method involves shifting a scaling diagonal matrix from left-multiplication to right-multiplication and merging it into the adjacency matrix. However it is a relatively common technique in numerical computing and compiler optimization. TopGQ does not introduce a fundamentally new approach to addressing the intrinsic challenges of GNN quantization, such as the highly non-uniform distribution of activations, as it does not alter the basic quantization paradigm but rather makes a more favorable quantization scheme (node-wise quantization) computationally feasible.

2. The two core contributions of the paper, i.e., "node-wise quantization" and "leveraging graph topology to guide quantization, are not novel concepts within the field. The studies such as A$^2$Q have already explored node-wise or mixed-precision quantization, and the use of centrality measures like node degree has long been established. The primary contribution of TopGQ lies in the straightforward combination of these two existing ideas, without substantial refinement of their underlying principles.

This paper lacks some necessary analysis.

1. The theoretical foundation of the paper (Theorem 1) relies on strong assumptions, such as Gaussian initialization, sufficiently large dimensions, and ReLU activation functions. However, the actual proposed TopPIN is a highly simplified first-order approximation. The transition from the intricate $\phi(v)$ to the simplified TopPIN lacks rigorous theoretical justification and ablation studies. This design choice is not sufficiently supported by analytical reasoning.

2. Does the effectiveness of TopGQ rely on the presence of a distinct topological structure that can be characterized by node degrees? In many real-world graphs (e.g., molecular or knowledge graphs), node features may play a far more critical role than topology, or the graph structure may be more complex than what degree alone can capture. Since DRA employs KL divergence for global range analysis and is applicable to various GNN architectures and datasets, does this make TopGQ's generality inferior to that of DRA?

The experimental design of the paper has some issues.

1. The comparison with relevant baselines is insufficient. When compared against numerous QAT methods, the observed speed advantage stems primarily from the inherent nature of post-training quantization (PTQ) being training-free, rather than from the novel contributions of this work. In terms of accuracy, TopGQ does not demonstrate a significant or consistent superiority over other methods.

2. The paper claims to achieve "orders of magnitude" speedup in quantization. To what extent is this acceleration attributable to the core concept of being training-free versus the specific implementations of TopPIN and dual-axis scale absorption? Shouldn't corresponding ablation studies be designed to isolate and validate the effectiveness of the proposed contributions?

There are obvious problems in the experimental analysis.

1. Figure 1 is not cited in the main text. Is the reference to "Section 1" in the INTRODUCTION section (i.e., in the last sentence of the first paragraph on Page 2) incorrect? Should it instead be referencing Figure 1?

2. The font size in some parts of Figure 3 is somewhat small. Increasing it would improve readability.

**Questions:**

1. Why was a first-order approximation chosen for TopPIN instead of a second or third-order approximation? Is this design choice supported by experimental evidence?

2. Does the effectiveness of TopGQ depend on the graph having a distinct topological structure that can be characterized by node degrees? How generalizable is the method to graphs with complex topological structures?

3. Does the effectiveness of TopGQ depend on the graph having a distinct topological structure that can be characterized by node degrees? How generalizable is the method to graphs with complex topological structures?

4. In the comparisons with DRA, under what conditions does TopGQ demonstrate superior performance, and in which scenarios does DRA hold the advantage?

---

### Note · Authors · 2025-11-24

**Comment:**

We sincerely thank the reviewers for their feedback and comments. In order to carefully address the concerns raised and allocate sufficient time for revision, we have decided to withdraw our submission. Nevertheless, we provide responses to the primary points raised by the reviewers below.
***

### **(2HBD, DDaU) 1. About the contribution of Scale Absorption and node-wise quantization**

$\to$ Our contribution is demonstrating that the benefits of node-wise scaling can be realized without actually executing node-wise quantization, which is typically infeasible on existing hardware. This addresses a long-standing gap between theoretically optimal quantization granularity (node-wise) and deployable quantization granularity (feature-wise), which the prior works haven’t accomplished.


### **(2HBD, DDaU, gi3j) 2. About the theoretical justification of TopPIN.**

$\to$  We note that the assumptions serve as abstractions to frame the link between topology and GNN activation, rather than strict requirements for applying TopPIN. Empirically, TopPIN’s consistent performance shows that this abstraction reliably reflects its objective in practice.


### **(DDaU) 3. The cited works discuss dynamic GNN training/inference, not quantization time bottlenecks. Thus, the paper’s claim that “quantization time is a deployment barrier” lacks direct evidence.**

$\to$ The cited works of dynamic GNNs are exactly the examples where long quantization time becomes a bottleneck, because an updated GNN would require a whole new quantization process for deployment. In such cases, the intolerable quantization time portrayed in Figure 1 would be frequently required.


### **(2HBD, gi3j) 4. Too few PTQ baselines make the comparison unfair.**

$\to$ There are no other concurrent PTQ methods for GNNs, other than DRA. Nearly all prior GNN quantization works are QAT-based. We believe this highlights the necessity of our method, a PTQ-based GNN quantization method.


***
Also, we provide the answers to other comments and questions as follows.

**(2HBD) Topology-guided quantization is not considered novel.**

$\to$ We respectfully clarify that the novelty of our method lies in linking graph topology to activation distribution and using it as a proxy for assigning quantization parameters. This enables the fastest GNN quantization pipeline and is fundamentally distinct from prior approaches.


**(2HBD) Where does the quantization time acceleration mainly come from?: being training-free vs. the proposed TopPIN and dual-axis scale absorption. The contribution of quantization speedup is from the nature of PTQ, not TopGQ.**

$\to$ It is from both. When compared to QAT baselines, the speedup mainly comes from being training-free. However, ours is much faster than even when compared to the PTQ baseline (DRA), which emphasizes the low computational cost of our method. The speedup is from eliminating gradient-based updates by leveraging topology.


**(2HBD) Design choice of TopPIN, why was a first-order approximation chosen for TopPIN instead of a second or third-order approximation?**

$\to$ Since TopPIN must be fast for unseen-node inference, the first-order variant provides the best balance of accuracy and efficiency. The higher-order terms offered similar accuracy gains while adding more computation.


**(2HBD, gi3j) Does the effectiveness of TopGQ depend on the graph having a distinct topological structure that can be characterized by node degrees? How generalizable is the method to graphs with complex topological structures?**

$\to$ Our method is grounded in local graph structure, which directly reflects the node-wise variance introduced during message aggregation. Thus, its effectiveness is maintained even as the global graph structure becomes increasingly complex.


**(2HBD) Under what conditions does TopGQ demonstrate superior performance, and in which scenarios does DRA hold the advantage?**

$\to$ TopGQ excels at low bitwidths, and especially when activations contain strong outliers, because it assigns per-node quantization parameters rather than a single global parameter like DRA. We note that per-node quantization with DRA is nontrivial, as it lacks an effective policy for assigning quantization parameters to unseen nodes at inference.


**(gi3j) Were the results of DRA reproduced by the authors or directly taken from the original publication?**

$\to$ The results were reproduced with our own implementation, as the official code was not disclosed.


**(gi3j) The claim “quantizing GNNs is known to be difficult due to the varying magnitudes of node activations” lacks empirical support.**

$\to$ The varying magnitudes of node activations are commonly mentioned in baseline methods [1,2], which motivates them to use node-wise quantization in the first place. Also, we show empirical support in Figure 2, where the node-wise activation range shows large variation between node indices.

[1] Tailor, S.A., et al. "Degree-Quant: Quantization-Aware Training for Graph Neural Networks." ICLR. 2021.

[2] Zhu, Zeyu, et al. "$\rm A^ 2Q$: Aggregation-Aware Quantization for Graph Neural Networks." ICLR. 2023.


**(gi3j) Lack of discussion about the improvements in Table 7.**

$\to$ We note that the detailed analysis regarding Table 7 is already provided in Section 6.5. Table 7 is the ablation study results of TopGQ.


**(gi3j) The selection of k from the k nearest neighbors.**

$\to$ We selected the best configuration by doing a hyperparameter search over $k=1-5$ for each experiment setting.


**(gi3j) The reported speed-up in Table 5 appears modest compared to the claimed efficiency gains.**

$\to$ Table 5 reports speedups at inference time, which is different from quantization time, where the order of magnitude speedup comes from. While TopGQ improves efficiency in both aspects, Table 5 only portrays one side of it.


**(gi3j) Discussion on the memory cost of TopGQ**

$\to$ The analysis about the memory cost is in the Appendix, Section G (Table 12, 13). Table 12 compares the theoretical storage cost between TopGQ and the baseline works, and Table 13 shows the overall memory cost benefits of using TopGQ, compared with FP32 inference.

**Withdrawal Confirmation:**

I have read and agree with the venue's withdrawal policy on behalf of myself and my co-authors.